# Morpho-Sedimentary Constraints in the Groundwater Dynamics of Low-Lying Coastal Area: The Southern Margin of the Venice Lagoon, Italy

Chiara Cavallina [1],*, Alessandro Bergamasco [2], Marta Cosma [1], Cristina Da Lio [1], Sandra Donnici [1], Cheng Tang [3], Luigi Tosi [1] and Luca Zaggia [1]

1   Institute of Geosciences and Earth Resources, National Research Council, 35131 Padova, Italy
2   Institute of Marine Sciences, National Research Council, 30122 Venice, Italy
3   Yantai Institute of Coastal Zone Research, Chinese Academy of Sciences, Yantai 264003, China
*   Correspondence: chiara.cavallina@igg.cnr.it; Tel.: +39-349-639-9065

**Abstract:** Complex freshwater–saltwater exchanges characterize most Holocene groundwater aquifers in low-lying coastal plains around the world, particularly in mechanically drained territories. This is due to the combination of several factors that control groundwater dynamics, including the high variability of the Holocene coastal deposits that host the shallow aquifers and the water management practices. The relationships between the stratal architecture of sedimentary deposits and the vertical changes in the salinity of the phreatic aquifer are poorly studied although they represent an issue of primary importance for a sustainable use of water resources and for agriculture. This research work is focused on the influence of sedimentary constraints, i.e., stratigraphic discontinuities and related changes in permeability in shaping salinity stratification into the unconfined aquifer at the southern margin of the Venice lagoon (Italy). Nine sites have been investigated by collecting sediment cores for facies analysis and monitoring water electrical conductivity in piezometric wells. The results show that buried channelized sandy deposits can enhance salinity mitigation of the phreatic aquifer in conjunction with precipitations and sufficient freshwater supply from nearby rivers and irrigation channels. Our analyses also reveal that the differences in stratigraphic architecture of the upper 10 m of the subsoil determine different fresh–saltwater dynamics of the phreatic aquifer. In particular, three possible behaviors can occur: (i) where the subsurface is characterized by the presence of a thick, up to 5 m, paleochannel, a freshwater lens is always present in the most surficial part of the phreatic aquifer; (ii) where the subsurface is composed by fine-grained sediments of marsh and lagoon paleo-environment, the phreatic aquifer tends to be salt-contaminated over its entire thickness; (iii) where the subsurface contains thin, up to 2–3 m, paleochannel deposits, the fresh–saltwater dynamics of the most surficial part of the phreatic aquifer varies more during the year, as a result of seasonal precipitation trend. The provided characterization of saltwater dynamics represents the basis for planning mitigation measures to improve the farmland productivity of the Venetian coastal plains.

**Keywords:** salt-water intrusion; groundwater dynamics; coastal plain

## 1. Introduction

Saltwater intrusion in shallow coastal aquifers has considerably increased over the past two decades because of sea-level rise and land subsidence along with increasingly long drought periods. Besides other economic activities, agriculture is the mostly impacted by the negative effects of saltwater intrusion, especially in low-lying farmlands [1,2], where measures are urgently needed to protect groundwater resources [3]. In this context, an integrated approach based on the knowledge of the multiple factors that affect the groundwater system is essential for a sustainable management of water resources. Among them, the stratigraphic architecture of the subsoil plays an important role in controlling

groundwater flow [4,5], and its comprehensive characterization is fundamental to understand the influence of stratigraphic constraints on salinization processes and to better control saltwater intrusion processes. In particular, the sequence stratigraphy approach, generally used for hydrocarbon exploration in ancient sedimentary successions, can be conveniently applied for predicting the continuity of aquifers and confining beds, with implications on the exploitation and management of underground freshwater resources (e.g., [6]). Campo et al. [7] applied this approach to the late Quaternary successions of the Po delta (Italy) to decipher complex spatial relations among aquifers. Detailed geomorphological and stratigraphic analyses, using facies analyses methods, are adopted in several studies in order to understand the groundwater flows and the stratigraphic architecture of aquifers in sedimentary deposits of floodplains and alluvial fans and deltas [8–12]. Some studies address the issue of aquifer heterogeneity, and thus the groundwater dynamics, through numerical models [13–17]. Others apply the stratigraphic analyses of sedimentary bodies to gas and oil reservoir systems [18,19]. Carol et al. [20] analyzed the role of geologic–geomorphic setting of the Holocene evolution of the coastal landscape of the Río de la Plata estuary to understand its control on the shape and distribution of freshwater lenses for water supply. Nevertheless, as coastal plains derived by the Holocene marine transgression display a high variability in the subsoil architecture even on a very small scale, investigations on stratigraphic controls on saltwater intrusion into these aquifer systems is still poorly developed in the literature, even if essential to address implications in the soil salinization process. Despite the numerous studies on the aquifer system of the coastal area of Venice, a comprehensive investigation on the role of stratigraphic features on the groundwater dynamics and on the saltwater–freshwater exchanges is still missing. Our research focuses on the uppermost subsoil, where the fresh–saltwater interface fluctuates and groundwater salinity can limit the growth of plants, with negative consequences on crop productivity [21].

The aim of our study is to highlight the morpho-sedimentary constraints in the groundwater dynamics of a salt-contaminated phreatic aquifer in the low-lying coastal plain south of the Venice lagoon. With the term "morpho-sedimentary constraint" we identify a stratigraphic discontinuity, causing a change in permeability, thus affecting groundwater dynamics. In this area, morpho-sedimentary constraints correspond to discontinuities between an upper permeable stratigraphic interval and a lower less permeable layer, confining at its top fresher and less dense water. The study analyzes how the presence of paleochannels into the phreatic aquifer influences mitigation processes of aquifer salinization, when there are enough freshwater inputs from precipitation or seepage from the channel bed of freshwater courses, in order to identify the areas most influenced by surficial freshwater availability and thus more sensitive to mitigation measures.

## 2. Study Area

The study area is located on the northern Adriatic Sea coast at the southern margin of the Venice lagoon (Italy), in a farmland territory just south of the lower Brenta and Bacchiglione rivers (Figure 1a–c). The coastal area surrounding the Venice Lagoon (Italy) (Figure 1) is a fragile environment subjected to both natural changes and anthropogenic pressures. This low-lying area experiences the combined effect of land subsidence and sea-level rise that boosts saltwater intrusion, inducing soil salinization and compromising agriculture [22–24]. During the last decades, several studies characterized the process of saltwater intrusion in this area. The first comprehensive study, carried out by Carbognin and Tosi [25], provided an insight on the saline plume dynamics. According to their results, the saline plume intrudes irregularly from the nearby sea and lagoon up to 20 km inland. Its top varies from 0 to 10 m below msl, whereas its bottom ranges between 15 and 70 m below msl, and locally reaches 100 m. De Franco et al. [26] detailed at local scale the seasonal saline dynamics by geophysical surveys at the southern lagoon margin. Gattacceca et al. 2009 and 2011 [27,28] provided the isotopic and geochemical characterization of the saline waters in the shallow aquifers. Viezzoli et al. [29], Teatini et al. [30] and Tosi et al. [31] analyzed

the exchanges between lagoon water and groundwater using airborne electromagnetic and marine continuous electrical resistivity tomography surveys. Da Lio at al. [32], and Tosi et al. [33] focused on the vulnerability of farmlands to saltwater intrusion in the Venetian coastal plain. Lovrinović at al. [34] reconstructed a model of the mechanism driving saltwater intrusion during wet and dry periods and high and low tide conditions, with superposed hydraulic reclamation activities.

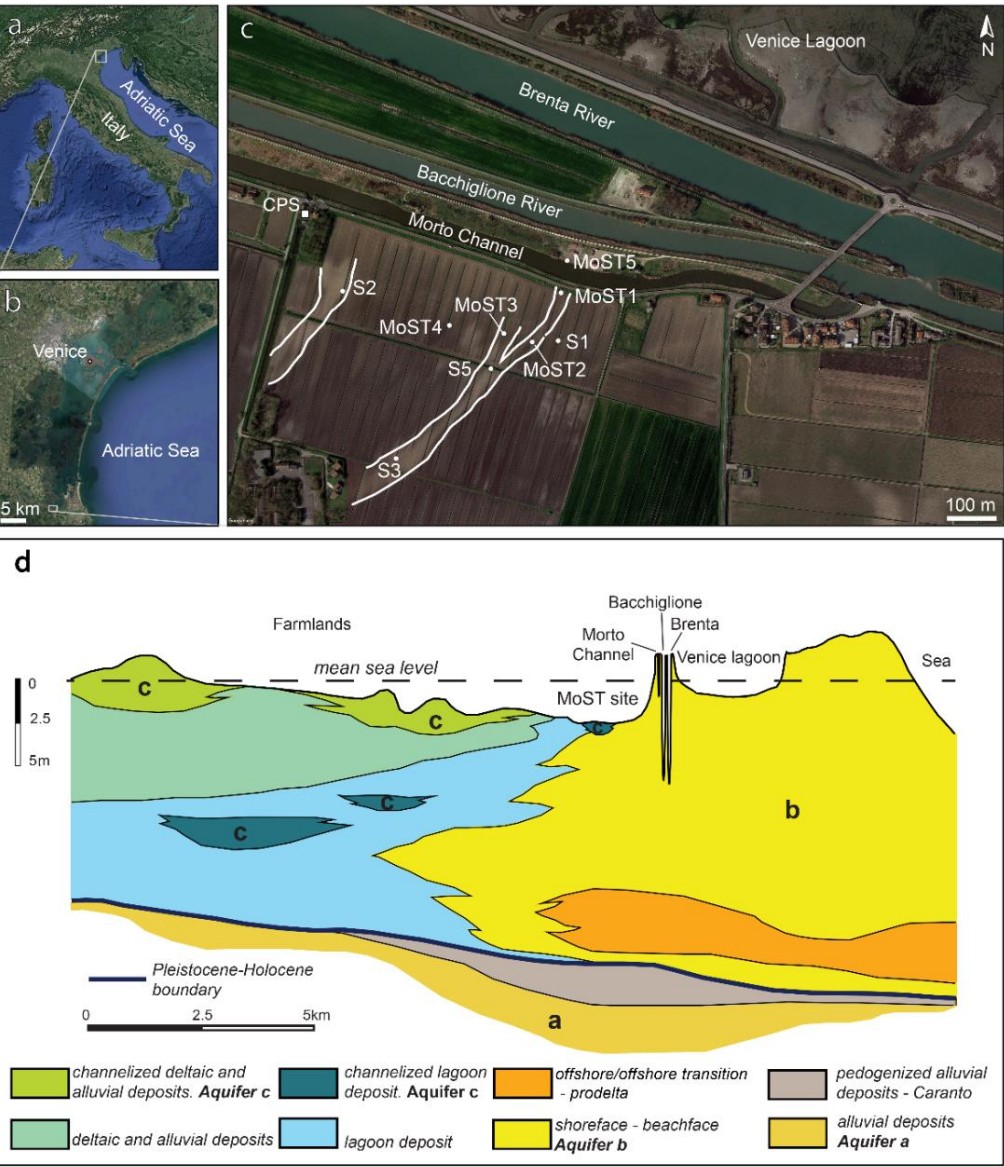

**Figure 1.** (**a**) Location of the study area along the coast of the Northern Adriatic Sea, Italy; (**b**) location of the study area at the southern margin of the Venice lagoon; (**c**) map of the Study area. Dots indicate the position of piezometric wells and cores (S1, S2, S3, S5, MoST1, MoST2, MoST3, MoST4, MoST5). White lines delineate boundaries of surficial sand bodies. CPS indicates the position of the Casetta Pumping Station. Base map source: Esri, Maxar, Earthstar Geographics, USDA FSA, USGS, Aerogrid, IGN, IGP and the GIS User Community; (**d**) stratigraphic setting of the Upper Pleistocene and Holocene succession at the southern margin of the Venice lagoon. Modified From [34].

The morphological setting of the study area is characterized by a low-lying territory, with ground elevation ranging between −3 m and +1 m from mean sea level. This altimetric difference is inherited from the elevation of ancient swamps, paleochannels and littoral ridges characterizing the area prior to the land reclamation [35]. Since the area mostly lies

below the mean sea level, the water table is artificially controlled by a pumping station (Casetta, CPS in Figure 1c) at a suitable level for irrigation and agricultural practices, discharging a collector, the Morto Channel, which drains the water surplus into the Brenta and Bacchiglione rivers through miter gates.

The regional stratigraphic architecture (upper 20 m of the subsoil) of the Venice coast, including the study area, consists of a sedimentary sequence composed by alluvial, transitional and marine late Pleistocene and Holocene deposits (Figure 1). Three main depositional phases are recorded corresponding to the global variations of the sea level during late Pleistocene and Holocene [36]: the Lowstand Systems Tract (LST), the Transgressive Systems Tract (TST) and the Highstand Systems Tract (HST). The late Pleistocene deposits (LST), consisting of alluvial sand, silt and clay, were accumulated during the Last Glacial Maximum (LGM) when the sea level was approximately 110–120 m lower than the present [37]. This continental succession ends with a paleosol, known as *caranto* [38,39], which represents a regional unconformity that marks the boundary between Pleistocene and Holocene deposits. The Holocene deposits, whose base dates back to 10–11 Kyr BP [40], show the typical wedge-shaped architecture of the transgressive (TST) and highstand (HST) sequences [36]. The TST sequence consists of lagoon/back-barrier deposits laterally passing to shallow marine deposits [41]. The following HST sequence is related to coastline progradation during sea level high stand and consists of fluvio-deltaic, lagoonal and littoral deposits, going from land to sea [40,42,43].

The hydrogeologic setting of the southern margin of the Venice lagoon is characterized by confined (a in Figure 1d), locally confined and unconfined aquifers (b and c in Figure 1d) outlining a complex aquifer system, which is a consequence of the complex lithostratigraphic architecture. The confined Aquifer a is hosted in the Pleistocene alluvial deposits and is bordered at its top by the *caranto* paleosol. The unconfined and locally confined ones (Aquifer b and Aquifer c), forming the phreatic aquifer of the area, are hosted in Holocene sandy deposits. In particular, Aquifer b is composed by littoral deposits, extending with continuity in the area, while Aquifer c is hosted in channelized morpho-sedimentary bodies, inherited by lagoon and fluvio-deltaic channels. In the subsoil of the study area (indicated as MoST site in Figure 1d), all the aquifer types are present with the superposition of local channelized Aquifers c directly on the Aquifer b. Due to the proximity of the sea and the lagoon, these shallow costal aquifers are affected by saltwater intrusion that extends irregularly up to 20 km inland. Groundwater salinization is driven by the superposition of different factors, such as the ground elevation of the area, the hydraulic drainage network, the seawater intrusion into the lower estuaries of Brenta, Bacchiglione, and the presence of preferential groundwater flow pathways along permeable buried sedimentary bodies as paleochannels and dune ridges.

## 3. Materials and Methods

In order to understand the relationships between stratigraphic and sedimentological characteristics of the subsoil, the hydrostratigraphic architecture and the groundwater dynamics, we followed a multidisciplinary approach, combining sedimentology and hydrogeology.

Nine sedimentary cores (MoST1, MoST2, MoST3, MoST4, MoST5, S1, S2, S3, S5) were described and analyzed in terms of physical features (lithology, grain size, sedimentary structures, color and sorting) and biological structures (presence of fossils, vegetal remains, bioturbation) identifying seven Facies Associations (FA), each one recording different paleoenvironments [44]. The FA have been related to Hydrostratigraphic Units (Hu) based on grain size (varying in the studied sections between clay and medium sand) and textures (i.e., sorting, presence of sedimentary structures), which reflect the permeability of the various parts of the aquifer system (see Table 1). Two perpendicular hydrostratigraphic sections across the study area, one oriented NNE-SSW (a-a′) and one W-E (b-b′), have been reconstructed, correlating the stratigraphy of the nine analyzed cores. The ground-

and well-head elevations were measured with respect to the mean sea level (msl) using a Differential Global Positioning System (DGPS).

**Table 1.** Correlation between the facies association and the hydrostratigraphic units as defined in this study and the seismic and sequence stratigraphic units defined in [36,41].

| Aquifer System unit | Facies Association | Hydro-Stratigraphic Unit | Seismic Units by [36,41] | Sequence Stratigraphy from [36,41] |
|---|---|---|---|---|
| Phreatic aquifer | Fa7 (4 m below msl—core top) | Hu7: Peat and peaty clay layers representing a thin discontinuous surficial impervious layer | H3: tidal channel and modern lagoonal deposits | Highstand systems tract |
| | Fa6 (7 m below msl—core top) | Hu6: Sandy deposits, locally silty, with high to medium permeability, representing local lenticular aquifer. | | |
| | Fa5 (12–4 m below msl) | Hu5: Sandy deposits with high permeability representing the littoral aquifer | H2b: Prograding delta front/prodelta, shoreface and beach ridge deposits | |
| | Fa4 (17–12 m below msl) | Hu4: Silty sediments with medium to low permeability representing a transition between very low aquifer permeability and aquitard | H1b: Transgressive shallow marine | Transgressive systems tract |
| | Fa3 (18–16 m below msl) | Hu3: Sandy–silty deposits with high to medium permeability representing a thin aquifer | H1a-H1b: Transgressive back-barrier deposits-Transgressive shallow marine | |
| Aquiclude | Fa2 (21–17 m below msl) | Hu2: Overconsolidated clay deposits with very low permeability representing an aquiclude | Pleistocene continental succession | Lowstand systems tract |
| Confined aquifer | Fa1 (core bottom—20 m below msl) | Hu1: Silty and sandy deposits with high to medium permeability representing an aquifer | | |

The phreatic aquifer was monitored 1 to 2 times per month from June 2020 to July 2021 through nine piezometers (MoST1, MoST2, MoST3, MoST4, MoST5, S1, S2, S3, S5) screened between 1 m and 10 m below ground level. The monitoring was carried out by deploying a CTD Diver®data logger in the piezometers and recording the electrical conductivity (EC), temperature (T) and pressure (P) every second during its lowering. Electrical Conductivity has been used an indirect indicator of the salinity of the groundwater of the aquifer. Piezometric head, meters on the mean sea level (msl), was obtained through barometric compensation of P data using atmospheric pressure recorded by BARO-Diver®and referred to the Italian geodetic datum. EC vertical profiles and hydro-stratigraphic sections have been referred to the msl and jointly analyzed in order to outline possible relationships between the relative positions of the salinity changes detected in the water column and the presence of sedimentary discontinuity revealed by detailed sedimentological and stratigraphical analysis. Finally, we selected the EC log of 9 October 2020 and 23 November

2020 as best and worst mitigation conditions, i.e., endmembers of maximum and minimum availability of freshwater. The two days have been selected in the same season to avoid major changes in water level due to the CPS activity. The recorded EC profiles have been interpolated along W-E and N-S sections in order to visualize the best and worst mitigation conditions in the studied period.

The morphosedimentary constraints are assumed to be the variations in the sedimentological features (lithology, grain-size, sorting) [45] influencing the subsoil permeability and capable to drive the salt–freshwater stratification, indirectly revealed by the EC values recorded in the aquifer. The presence of a morphosedimentary constraint corresponds to points of maximum variation on the EC curve, where lower values of EC start to increase, indicating a less permeable layer confining fresher and consequently less dense water above its top.

## 4. Results

### 4.1. Facies Associations and Hydrostratigraphic Units

Sedimentological analyses performed on the nine sediment cores allow the recognition of seven sedimentary facies associations from which, based on their hydrogeological behavior, an equivalent number of hydrostratigraphic units were identified (See Figures 2 and 3 and Table 1).

**Confined Aquifer**

- Facies association one (FA1) exceeds 3 m in thickness and occurs in the lowest part of the study cores. It is formed by brownish–grayish sandy silt, locally presenting sub-horizontal to inclined millimetric lamination, vegetal remains and roots. These sedimentological features, together with data from previous studies [36,40,41,46] allow for attributing FA1 to the alluvial environment occurred in the area during the Last Glacial Maximum. From a hydrogeologic perspective, this facies association (HU1) represents the uppermost surficial confined aquifer (described in literature) [40,41,47] hosted in the Pleistocene continental deposits and confined at the top by the *caranto* paleosol.

**Aquitard**

- Facies association two (FA2) is 3–4 m thick and consists of light brown clay to silty clay layers showing evident signs of pedogenesis, such as mottling and caliche nodules. The top of FA2 is marked by a bioturbated erosional surface. This facies association refers to the over-consolidated *caranto* paleosol. Being composed of fine-grained sediments, the hydro-stratigraphic unit associated to FA2 corresponds to an aquiclude (HU2), confining at the top the aquifer represented by HU1.

**Phreatic Aquifer**

- Facies association three (FA3), showing a thickness of about 1 m, unconformably overlies through an erosional surface FA2, and it is composed by poorly sorted, grey silt to silty sand structureless deposits. In the lower part, it contains 2 or 3 cm thick layers of medium to coarse sand showing a basal erosional surface with abundant shell fragments and shells of bivalves and gastropods. Coarse-grained sediments and shell lags suggest a high-energy coastal environment. The erosion surfaces are interpreted to be the result of wave ravinement that cut underlying deposits, in response to rapid beach–barrier migration [48,49]. The shell-rich layers correspond to a transgressive lag, and the overlying sand is attributed to the back-barrier to the shoreface environment. This facies association corresponds to a thin aquifer (HU3), which is the deepest part of the phreatic aquifer system of the area [40,41,46].
- Facies association four (FA4) is 3.5–5 m thick and gradually overlies FA3. It is composed by an alternation of millimetric to centimetric thick layers of grey silt and sandy silt. It shows planar to cross laminations, rare vegetal remains, cm-thick sandy layers with marine bivalves, gastropods and shell fragments, testifying the occurrence of sediment transport and deposition under variable energy such as that developing in an offshore/offshore-transitional, possibly a prodelta, environment [40,41]. Being that

FA4 is mostly composed by silty sediments with medium to low permeability, the related hydro-stratigraphic unit four (HU4) represents a transition between very low aquifer permeability and aquitard.

- Facies association five (FA5) ranges in thickness between 8 and 10 m. It consists of yellowish-gray structureless sand with a fossiliferous layer at the base. Very well sorted, fine to medium sand, with abundant shells and shell fragments, dominates at the base, while fine to medium sand, moderately sorted, with rare shell fragments and vegetal remains dominates in the upper portion. In the middle portion of this facies association are locally present 1–2 m of silty sand to silty layers with vegetal remains and shells. FA5 is interpreted to be developed in shoreface to beachface environment [47]. The hydro-stratigraphic unit related to FA5 (HU5), being composed by sandy permeable deposits, represents a major part of the phreatic aquifer of the area [46,47].

- Facies association six (FA6) is laterally discontinuous and erosionally overlies Fa5. Its thickness ranges between 5 and 0.5 m. FA6 contains grey to yellow moderately sorted medium to fine sands, alternating with sandy, silty clay and clayey-silt horizontal laminae, locally showing shell fragments or vegetal remains at their bases. The upper portion of FA6 shows mottling and traces of oxidation. The base of FA6 depicts a lenticular geometry, often marked by the presence of a fine layer containing abundant vegetal remains. These sedimentological characteristics, together with the mollusk shells and foraminifera content, indicate a lagoon-littoral transitional environment characterized by active exchanges with the sea. FA6 refers to the infilling of lagoon paleochannels, whose lenticular shapes are often well recognizable by remote sensing imagery (Figure 1). The related hydro-stratigraphic unit (HU6) represents local thin aquifer lenses contained in the lenticular sandbodies. The base of this aquifer is often composed by silty sediments, separating it from the littoral sands of HU5.

- Facies association seven (FA7) refers to discontinuous deposits of the uppermost subsoil layer, immediately below the arable land. It shows a maximum thickness of 4 m. FA7 contains peat, peaty-clay layers or clay levels with abundant vegetal remains of the wetlands existing in the area before the hydraulic reclamation that was performed at the beginning of 20th century. The reduced thickness or the absence of these uppermost deposits is due to the subaerial oxidation of peats and the related geochemical land subsidence. Because of its low permeability, the related hydro-stratigraphic unit (HU7) is considered an impervious layer.

### 4.2. Hydrostratigraphic Model

A hydro-stratigraphic reconstruction of the study area was obtained by correlating the sedimentary facies associations and hydro-stratigraphic units found in the nine cores supported by the concepts of sequence stratigraphy, whose models for the Venice area are available from previous studies (See Table 1 for the correlation between FA, HU and seismic and sequence stratigraphic units defined in previous studies) [36,40–42,50]. The hydro-stratigraphic model reconstructed for the study area is depicted by two sections oriented N-S and W-E in Figure 3.

### 4.3. Groundwater Dynamics
#### 4.3.1. W-E Section

- **S2:** The EC values recorded in the water column of well S2 show a strong variability over the year, ranging between 2 and 30 mS/cm. The EC vertical profiles display the presence of a persistent freshwater lens floating over the salt-contaminated aquifer. The thickness of the freshwater lens varies from a few decimeters to about 5 m. Four main possible configurations of the fresh–saltwater interface have been recorded (Figure 4): (i) In the first configuration, the major change in EC is located at about 7 m below msl, in correspondence with a thin layer of silt inside the channelized aquifer (HU6). In one case, recorded on 11 November 2020, the aquifer presents 2–3 mS/cm down to

this level; in the other two cases, recorded in Spring 2020, freshwater is present only in the first centimeters of the aquifer corresponding to the base of the agricultural soil; then the salinity rapidly increases, reaching 17–18 mS/cm at 4 m depth, and remains constant down to 7 m below msl, where the silty layer occurs in HU6. Below this level, the salinity rapidly increases, reaching its maximum value (28 mS/cm), and remains constant down to the bottom of the aquifer (Figure 4a). (ii) The second configuration is defined by a thicker freshwater lens (with EC values up to 5 mS/cm), which reaches the base of the channelized aquifer (HU6), at around 8 m below msl, where thin layers of silt occur (b). (iii) The third configuration is identified by a freshwater lens that reaches its maximum thickness down to 9 m below msl where a decrease in the grain size occurs within the sandy littoral aquifer (HU5) (Figure 4c). (iv) In the last configuration, the fresh–saltwater interface gradually deepens, representing transitional steps in the mitigation of the salt-contaminated aquifer, depending on the availability of freshwater (Figure 4d). Summarizing, the sedimentological constraints of the groundwater dynamics in well S2 lie at, 7 m, 8 m and 9 m below msl, influencing the depth of fresh–saltwater interface depending on the availability of freshwater in the aquifer. A freshwater lens is always present in this part of the aquifer, varying its depth roughly between 3.5 and 9 m below msl (Figure 4e).

- **MoST4:** The vertical EC profiles recorded in MoST4 well present values between 20 mS/cm and 30 mS/cm, from the top to the bottom of the aquifer. The EC does not show a strong vertical variability. A change in EC values is shown at the very top of the aquifer, where the curve shows a deflection to lower values (down to 20 mS/cm, Figure 5), in correspondence of a peaty clay layer at the top of the subsoil (HU7). The rest of the vertical profiles do not display other evidence of changes in salinity. The subsoil in this area is quite homogeneously composed by sand of the littoral aquifer (HU5). One stratigraphic constraint to the groundwater dynamics is at around 4 m below the msl, where a peat layer occurs (HU7) corresponding to the lower levels of EC at the top.

- **MoST3:** The EC values recorded in the water column of MoST3 well range between 18 mS/cm and 35 mS/cm. The EC shows a strong vertical variability identifying two possible configurations (Figure 6) of the fresh–saltwater interface through the year: (i) the first 2 m of the aquifer present a water lens with EC values variable between 18 and 30 mS/cm. The base of the lens is confined by a thin silty layer at the base of the channelized aquifer (HU6) at around 6 m below msl. Below this level EC values are higher varying between 28 and 32 mS/cm; (ii) the other profiles oscillate between 30 and 35 mS/cm without a vertical stratification. Summarizing, the stratigraphic constraint in MoST3 lies at 6 m below msl, periodically hosting a water lens whose EC values depend on the availability of freshwater.

- **MoST2:** The EC profiles recorded in the groundwater of MoST2 well show a large vertical variability, between 2 and 42 mS/cm, while there is no significant change over the year except for the very upper part. The aquifer presents two possible configurations of the vertical EC curve: (i) lower EC values, between 2 and 5 mS/cm, are recorded in the first meter, down to 4 m below msl, corresponding to a thin layer of silty clay at the base of the channelized aquifer (HU6). Below this level, the salinity rapidly increases, reaching EC values of 35–38 mS/cm. These values remain constant down to around 8.5 m below msl. At this depth, corresponding to layers of silt and peaty clays in the littoral aquifer (HU5), the EC values increase again, reaching 40–42 mS/cm (Figure 7a) (ii) The major changes in salinity are located in the same position of the first configuration, but the EC values of the first meter never get below 17 mS/cm (Figure 7b). Summarizing the EC profiles outline the presence of two sedimentological constraints: the upper one (4 m below msl) corresponds to the base of the HU2, and the lower one corresponds to a thin clay layer in the littoral aquifer

(HU5) at about 9 m in depth. The salinity of the first meter of the aquifer varies during the year, depending on the availability of freshwater (Figure 7c).

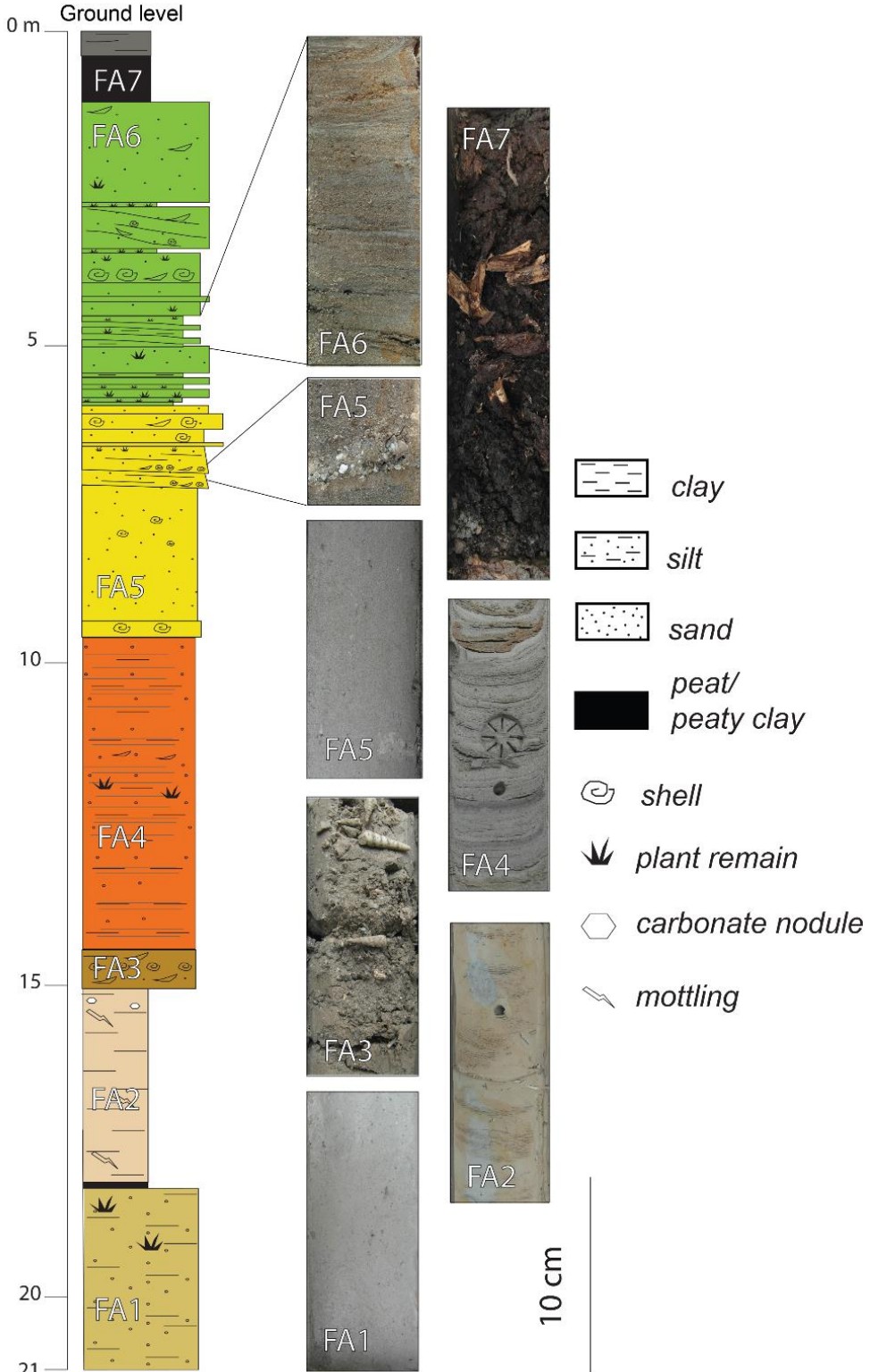

**Figure 2.** Stratigraphic column representative of the studied succession and images of each facies association described in the text. The section schematizes the general stratigraphy of the area, and it is not linked to any particular core. The 10 cm scale refers to the core pictures on the right of the figure.

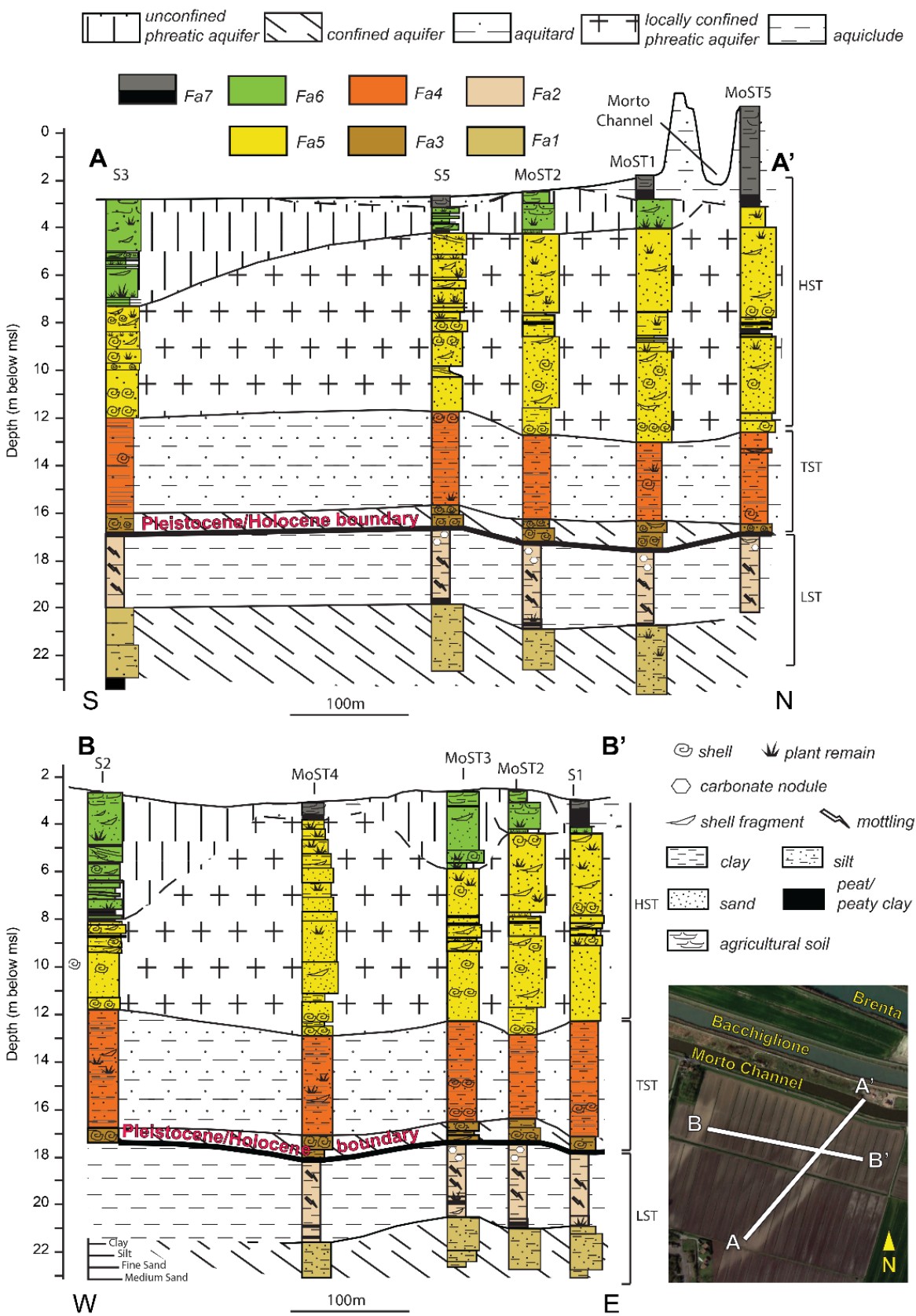

**Figure 3.** Hydro-stratigraphic sections of the study area (A-A′ and B-B′ in the inset map).

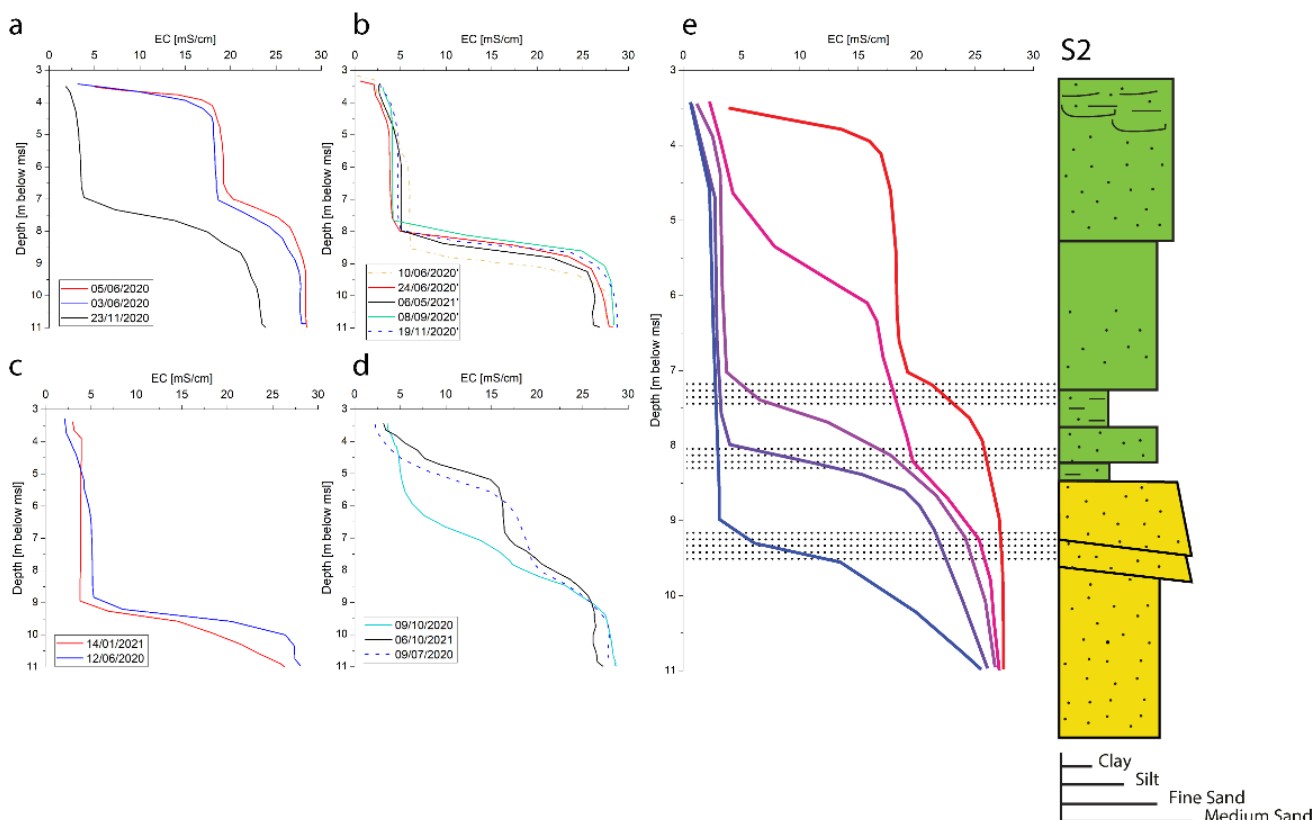

**Figure 4.** EC profiles collected in S2 well. Panels (**a–d**) show the 4 configurations assumed by the EC curve during the observed period. The model in panel (**e**) summarizes the configurations and compares them with the stratigraphic section, outlining the sedimentary constraints (dotted areas) existing in the subsoil surrounding the monitoring point.

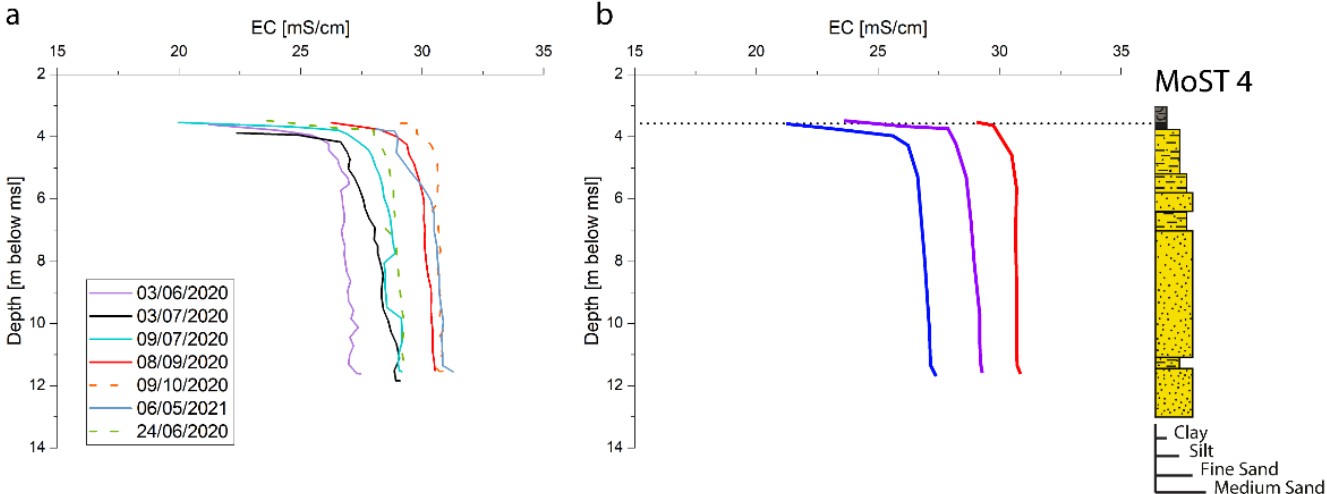

**Figure 5.** EC profiles collected in MoST4 well. Panel (**a**) shows the configurations assumed by the EC curve during the observed period. The model of panel (**b**) summarizes the configurations and compares them with the stratigraphic section, outlining the sedimentary constraint (dotted area) existing in the subsoil surrounding the monitoring point.

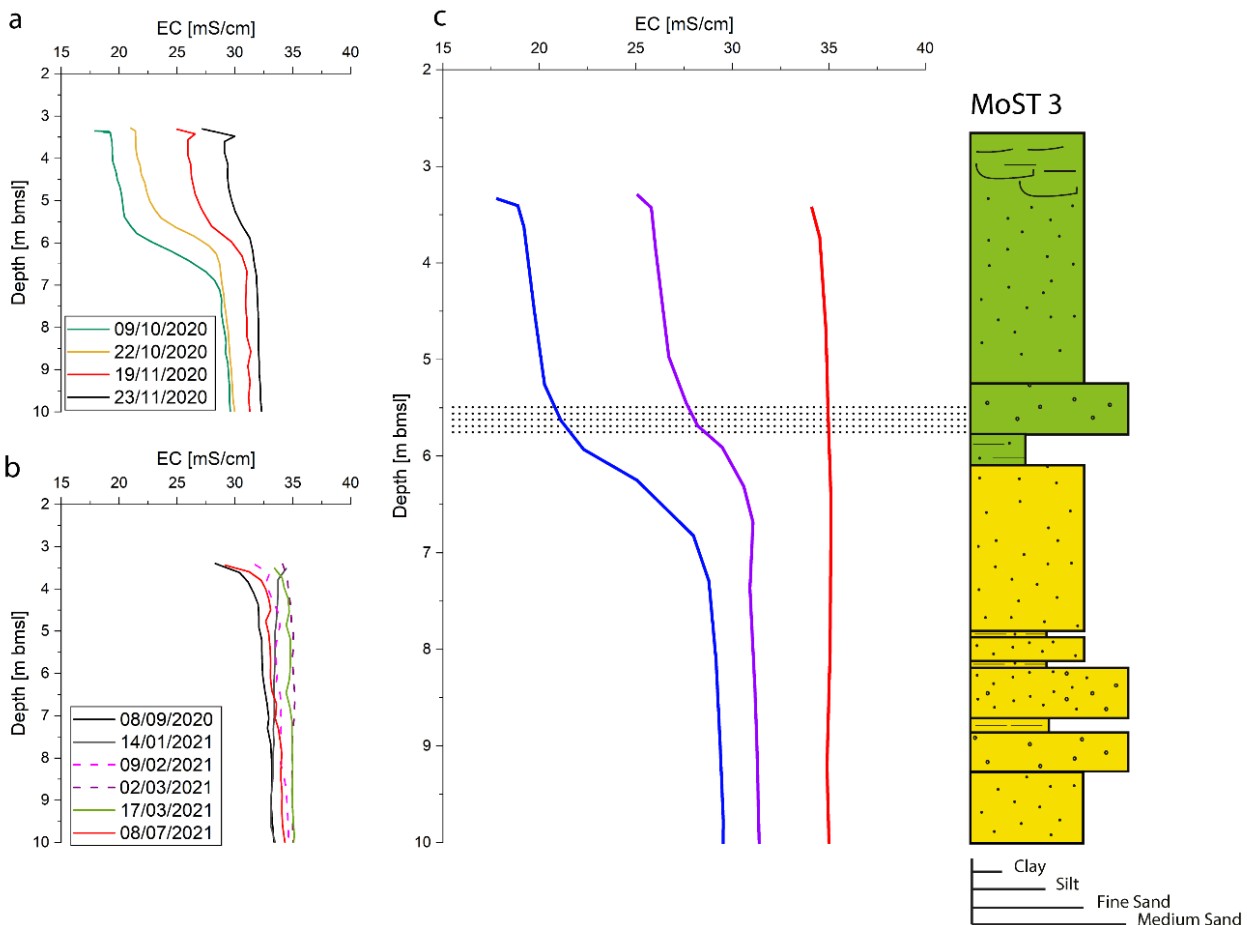

**Figure 6.** EC profiles collected in MoST3 well. Panels (**a**,**b**) show the configurations assumed by the EC curve during the observed period. The model of panel (**c**) summarizes the configurations and compares them with the stratigraphic section, outlining the sedimentary constraint (dotted area) existing in the subsoil surrounding the monitoring point.

- **S1:** The vertical EC profiles recorded in the easternmost well (S1) display moderate vertical variability, with values ranging between 35 and 45 mS/cm, and low variability over the year, presenting two configurations (Figure 8): (i) the EC values vary between 35 and 42 down to about 8 m below msl, corresponding to a clayey silt layer in the littoral aquifer (HU5). Below this level the EC values reach 45 mS/cm and remain constant down to the bottom of the aquifer; (ii) the variability of salinity is very low, with values between 40 and 45 mS/cm throughout the whole thickness of the aquifer. Summarizing, even if the EC curve presents high values from the top to the bottom of the aquifer, the slight change in EC values in configuration (i) suggests the presence of a sedimentological constraint at around 8 m below msl. Above this level, the salinity of the aquifer could present slightly lower values depending on the availability of freshwater.

4.3.2. N-S Section

- **MoST5**: The northernmost well of the study area (MoST5) is located at the northern bank of the Morto Channel, and it is the only well of the studied network located over msl (1 m over msl). The vertical EC profiles present extremely low variability, being always fresh with values around 0.5–3 mS/cm from the top to the bottom, due to the seepage of freshwater from the Morto Channel. The recorded profiles (Figure 9a) show a small increase in the EC, from a minimum of 0.5 to a maximum of 3 mS/cm, at 3 m

below msl, corresponding to a peaty clay layer at the base of HU1. Below the clay layer, the value remains constant, between 2 and 3 mS/cm, along the whole profile. The only stratigraphic constraint in this profile is located at 3 m below msl. Since the water is fresh and not stratified, no other sedimentological constraints can be found/identified.

- **MoST1**: The EC recorded in MoST1 well indicates a very high vertical variability between 0 to 50 mS/cm in the upper 3 m and different behaviors over the year. Four different configurations of the fresh–saltwater interface are present (Figure 10): (i) in the first configuration a freshwater lens, 3 m thick, lays on a thin silty layer at 5 m below msl, corresponding to the base of the local channelized aquifer (HU6). Below this level, the salinity rapidly increases, reaching values between 33 and 40 mS/cm; (ii) the second configuration does not show a vertical stratification in the upper part, presenting an EC value of about 40 mS/cm; (iii) and (iv) configurations represent intermediate conditions in the fresh–saltwater dynamics of the upper part of the aquifer, possibly also influenced by the presence of minor sedimentological constraints: a thin silty layer inside HU2, for the third one, and a surficial peat layer at 2–2.5 m for the fourth one. In all the configurations, the EC profiles show a change at about 9 m below msl, where thin fine layers occur inside the littoral aquifer (HU5). Below this level the EC reaches its maximum value (45–48 mS/cm). Summarizing, two main sedimentological constraints were identified at around 5 and 9 m below msl. In the first 3 m of the aquifer, a freshwater lens is often present, with variable thickness depending on the freshwater availability.

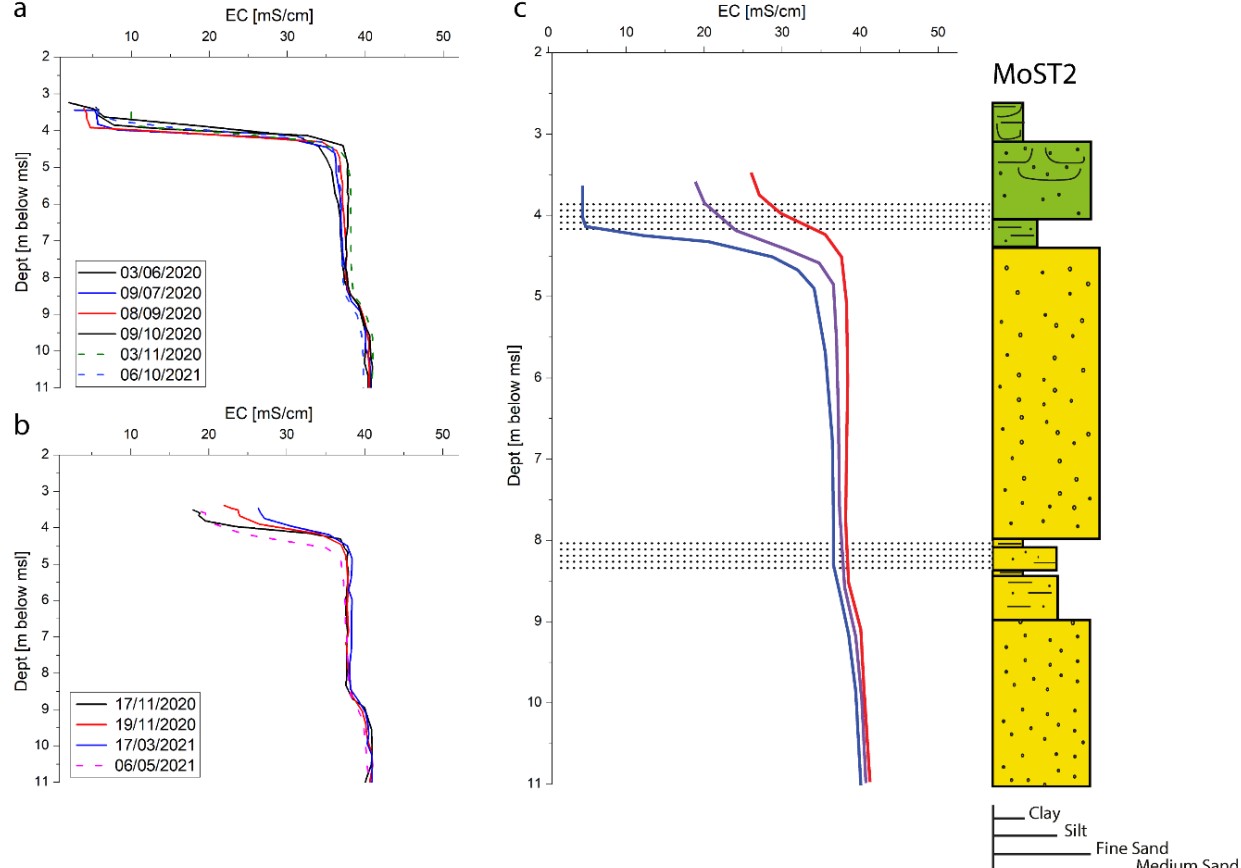

**Figure 7.** EC profiles collected in MoST2 well. Panels (**a**,**b**) show the configurations assumed by the EC curve during the observed period. The model of panel (**c**) summarizes the configurations and compares them with the stratigraphic section, outlining the sedimentary constraints (dotted areas) existing in the subsoil surrounding the monitoring point.

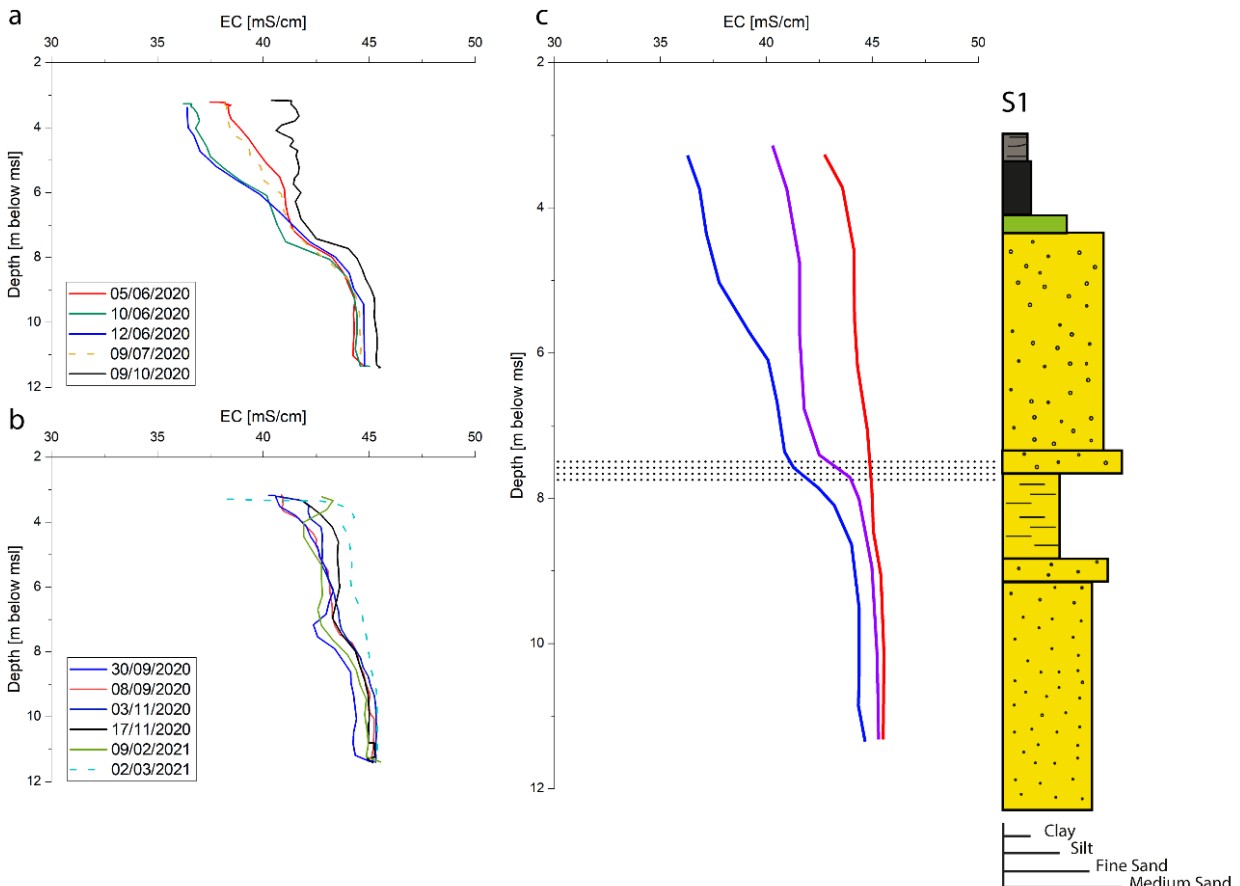

**Figure 8.** EC profiles collected in S1 well. Panels (**a**,**b**) show the configurations assumed by the EC curve during the observed period. The model of panel (**c**) summarizes the configurations and compares them with the stratigraphic section, outlining the sedimentary constraint (dotted area) existing in the subsoil surrounding the monitoring point.

- **MoST2:** see in the previous section
- **S5**: The EC recorded in the S5 well shows low vertical and seasonal variability, with values ranging between 21 and 27 mS/cm. Two configurations are recognized (Figure 11): (i) in the first configuration, the first 1–1.5 m, corresponding to the channelized aquifer (HU6) presents lower EC values. Below 5 m depth, the profiles remain constant around 26 mS/cm; (ii) the other EC recorded in S5 well present a vertical constant profile, with values around 26 mS/cm. Summarizing, the fine layer at the base of the channelized aquifer (HU6) represents the only stratigraphic constraint of the groundwater dynamics in this portion of the aquifer.
- **S3**: The EC profiles recorded in S3 well presents a high vertical variability, with values between 0 to 25 mS/cm (Figure 12). The lower portion is constantly between 20 and 25 mS/cm, while the upper 6–7 m present a strong variability over the year, with a surficial freshwater lens confined between the base of the agricultural soil and silty layers inside the local channelized aquifer (HU6, Figure 1). Within this range, the fresh–saltwater interface can assume intermediate configurations with fresher water lens with various thicknesses over the year. Below 2 m depth (5 m below msl), the EC rapidly increases reaching 20–25 mS/cm at the top of the littoral aquifer (HU5). Roughly between 5 and 7 m below msl, the groundwater can present different salinities, depending on the availability of freshwater, never exceeding 20–22 mS/cm. Another change in EC is found at about 7 m below msl, where a silty layer at the base of HU2 confines relatively fresher water. Below this level, the EC reaches its

maximum values, between 22 and 25 mS/cm, and remains constant down to the bottom of the aquifer. Summarizing, the phreatic aquifer in this area present two main stratigraphic constraints, at 5 and 7 m below msl, respectively, corresponding to fine sedimentological layers inside the HU6.

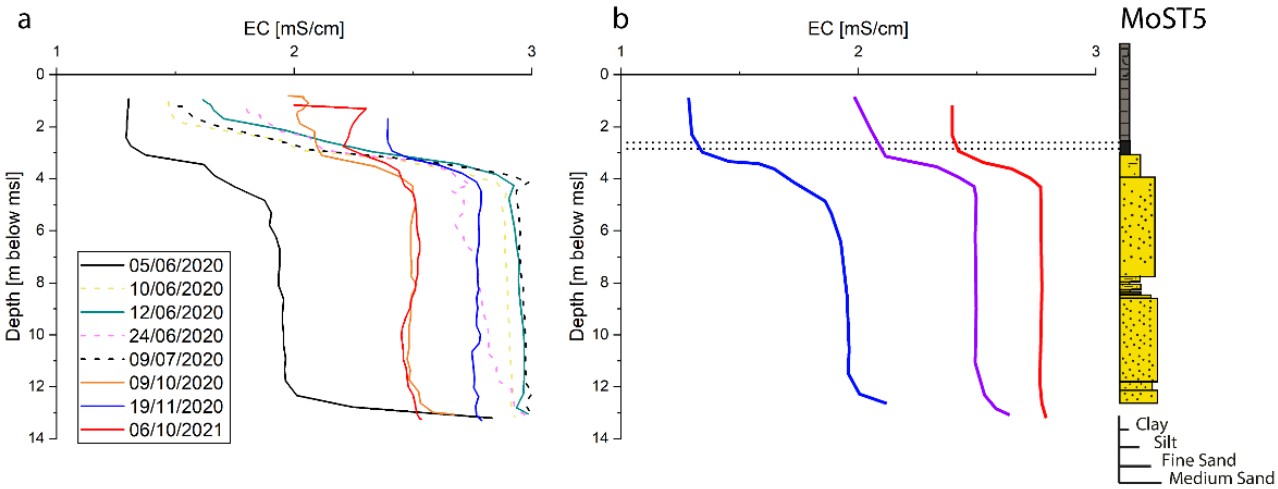

**Figure 9.** EC profiles collected in MoST5 well. Panel (**a**) shows the configurations assumed by the EC curve during the observed period. The model of panel (**b**) summarizes the configurations and compares them with the stratigraphic section, outlining the sedimentary constraint (dotted area) existing in the subsoil surrounding the monitoring point.

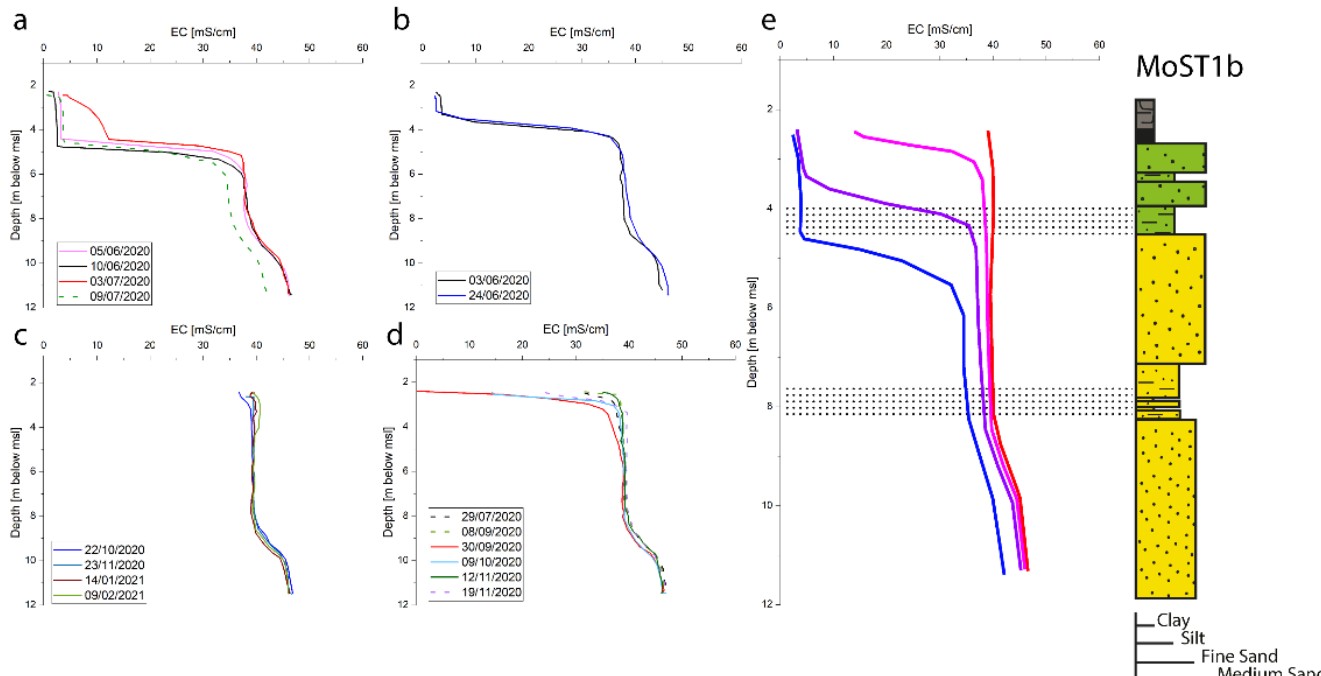

**Figure 10.** EC profiles collected in MoST1b well. Panels (**a**–**d**) show the configurations assumed by the EC curve during the observed period. The model of panel (**e**) summarizes the configurations and compares them with the stratigraphic section, outlining the sedimentary constraints (dotted areas) existing in the subsoil surrounding the monitoring point.

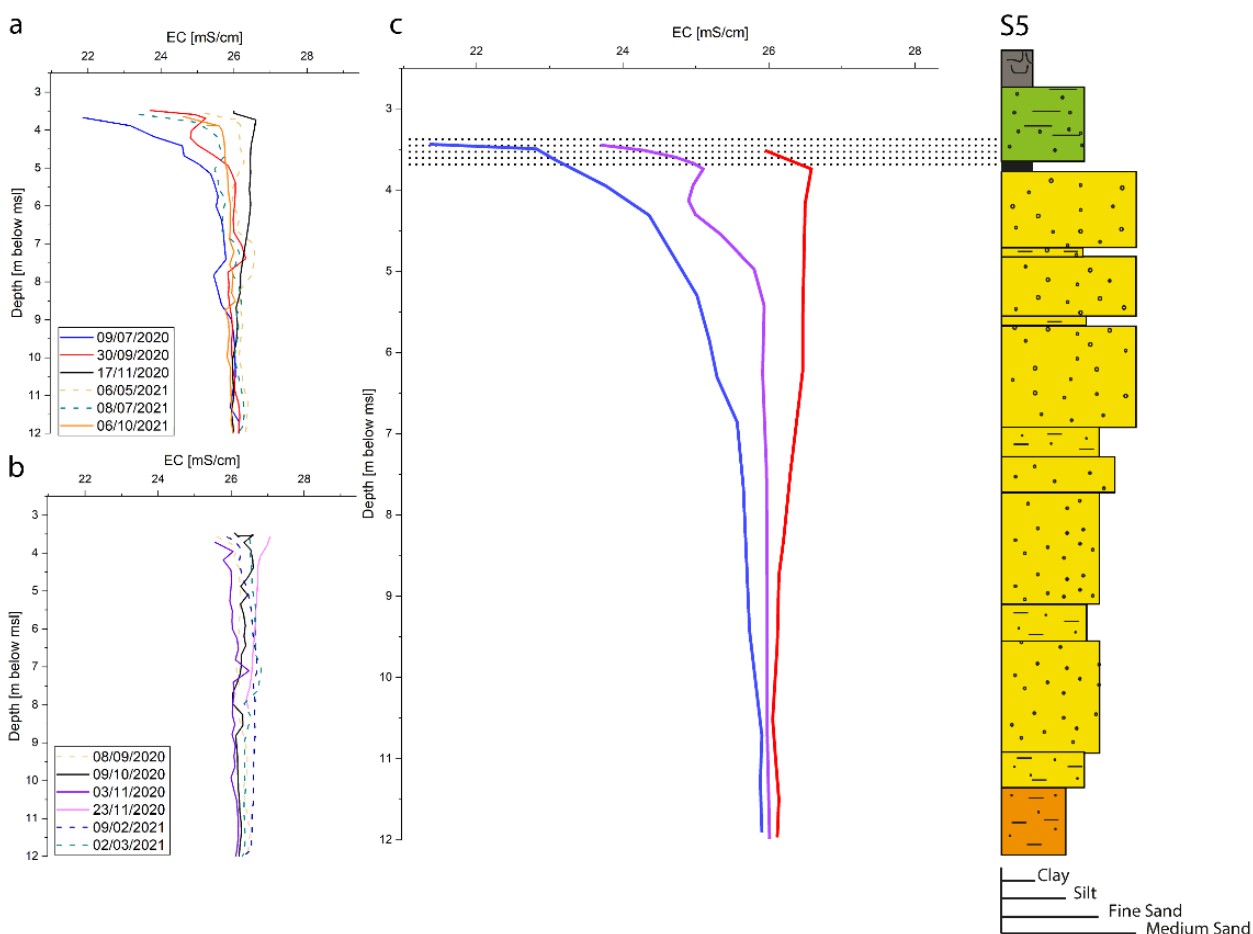

**Figure 11.** EC profiles collected in S5 well. Panels (**a**,**b**) show the configurations assumed by the EC curve during the observed period. The model of panel (**c**) summarizes the configurations and compares them with the stratigraphic section, outlining the sedimentary constraint (dotted area) existing in the subsoil surrounding the monitoring point.

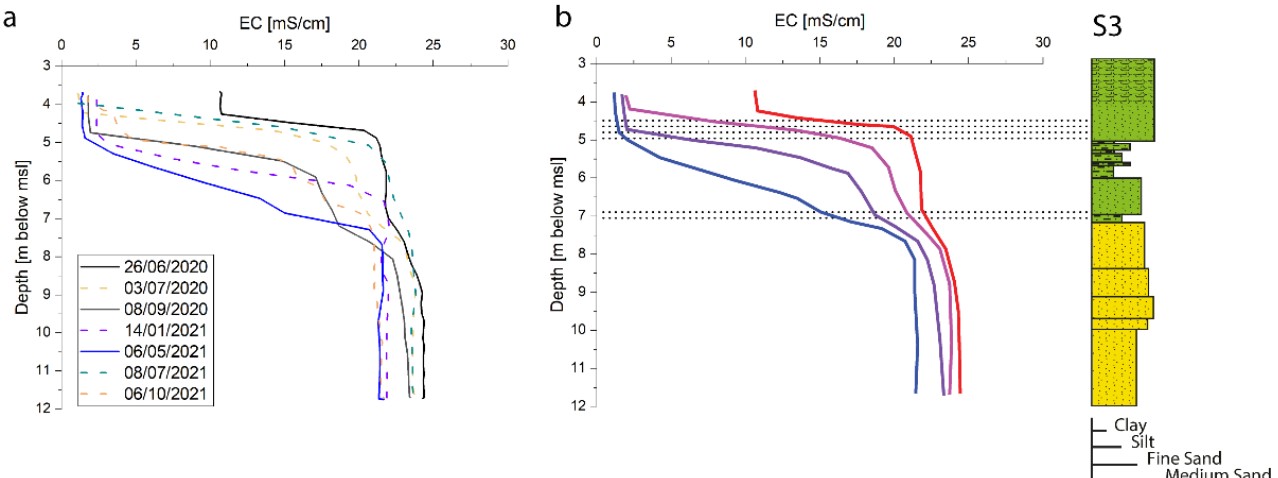

**Figure 12.** EC profiles collected in S3 well. Panel (**a**) shows the configurations assumed by the EC curve during the observed period. The model of panel (**b**) summarizes the configurations and compares them with the stratigraphic section, outlining the sedimentary constraints (dotted areas) existing in the subsoil surrounding the monitoring point.

## 5. Discussion

The fresh–saltwater stratification in groundwater is generally controlled by density differences. However, sedimentary constraints corresponding to local stratigraphic–lithological discontinuities determine differences in the permeability of the aquifer without confining the aquifer itself. The presence of finer sand or thin silty layers interbedded to medium-coarse sand can exacerbate the separation between water with density differences, despite their slight difference in salinity. This process is well evidenced when direct precipitation or river waters recharge the aquifer. Indeed, significant changes in the shapes of the EC profiles are better observed where thin lenses of fresher water in the upper part of the aquifer occur.

### 5.1. Fresh–Saltwater Dynamic Model

In general, the analyses of the various wells of the studied area have shown that small sedimentological discontinuities, such as the presence of fine-grained layers, although not effectively confining the aquifer, can significantly affect groundwater stratification by controlling the shape and location of the interface and the transition zone formed between freshwater and saltwater. Based on this information, and considering that the major freshwater recharge of the phreatic aquifer is related to seepage from the channel bed of rivers and irrigation channels and direct precipitation, a conceptual model has been outlined (Figure 13). The model shows how the stratigraphic architecture of the subsoil could modify the vertical profile of the EC, assumed as representative of the groundwater density (i.e., the salinity) and highlights that the EC profile shape (i.e., the groundwater stratification) can be strongly influenced by the presence of stratigraphic discontinuities. Two cases based on different stratigraphic setting are considered: a sandy homogeneous subsoil and a subsoil containing a silt layer that interrupts the stratigraphy. Both cases have been conceptualized considering an increasing availability of freshwater. The model show how the EC curve varies between two end-members: no input, on the left, and large input of freshwater from the top of the aquifer, on the right.

In the case of a homogeneous subsoil, almost constant values of EC represent vertical profiles under both negligible and high freshwater inputs. Between these two extreme EC conditions, the progressive increase in freshwater in the uppermost part of the aquifer allows to form a transitional zone where the salinity gradually increases towards the bottom of the aquifer.

In the case of a stratified subsoil, a vertical EC profile forms, as in the case of the homogeneous subsoil, under high salinity conditions and the absence of freshwater in the upper part the aquifer. Conversely, the EC profile shows a sharp change in correspondence of the fine-grained layer when the freshwater input progressively increases in the upper part of the aquifer. The fine-grained layer acts as a constraint of the groundwater salinity dynamics that delays the infiltration of freshwater into the deeper part of the aquifer. The freshening signal hardly propagates vertically, passing through the fine and less permeable layer, and preferably expands laterally with the progressive increase in extension and freshness of the surficial lens.

### 5.2. Stratigraphic Constraints in N-S and W-E Sections

Five setting where stratigraphic constraints occur have been recognized in two orthogonal sections crossing the study area (Figure 14):

- At the base of Hu6, where clay and silty layers are present. The presence of the channelized aquifers triggers the formation and the maintenance through time of fresher water lenses in the phreatic aquifer. This occurs homogeneously along the N-S section, corresponding to a N-S directed paleochannel, and in correspondence of MoST2, MoST3 and S2 points, in the W-E section.
- At the top of clay and silty layers in Hu6. This occurs in the western channelized surficial sandbody (observed in S2 point) and in the southern part of the eastern

one (observed in S3 point), where the channelized aquifers (Hu6) show their major thickness.
- In the upper part of the littoral aquifer Hu5 where coarser sand grades downward to finer sand (as in S2 point).
- Around the middle portion of Hu5, where silty-clay layers occur (in S1, MoST1 and MoST2 points).
- At the base of Hu7, in correspondence to peaty clay layers that could preserve fresher water at the very top of the phreatic aquifer (as it was observed in MoST5 and MoST4 points).

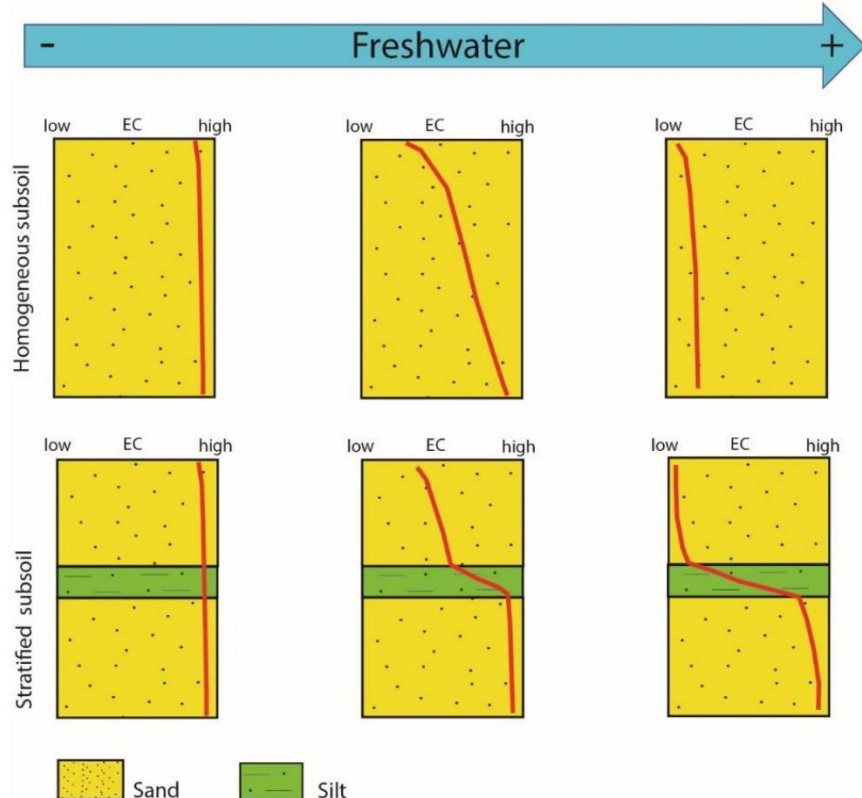

**Figure 13.** Conceptual model of the propagation of the mitigation signal in surficial aquifer characterized by homogeneous or stratified subsoil. The salinity of the groundwater is represented by the EC parameter.

*5.3. Fresh–Saltwater Dynamics in Best and Worst Mitigation Condition*

Best and worst mitigation condition of the fresh–saltwater dynamics refer to 9 October 2020 and 23 November 2020, respectively.

5.3.1. N-S Section (Figure 15a)
- Best mitigation conditions: the EC of the aquifer is characterized by the presence of an extended salty bulge in the middle part of the area, recorded in MoST1 and MoST2, featuring an EC higher than 35 mS/cm and reaching the depth of around 5 m below sea level. The top of the aquifer, in this area, is characterized by the presence of a fresher water lens up to 1–2 m. To the north, the EC strongly decreases, and the aquifer results fresh from the bottom to the top (MoST5), due to the presence of the Morto Channel, whose freshwater seeping from the channel bed infiltrates into the subsoil. To the south, the deep portion of the aquifer is still salty, around 20–25 mS/cm, while it freshens towards the top, where a freshwater lens up to a depth of 5 m is present (S2).
- Worst mitigation conditions: the saltwater bulge in the middle portion of the aquifer reaches the surface, the fresher water lens disappears. The freshwater lens in the

southern portion of the area (S3) is still present, even if it is smaller and the EC values are higher.

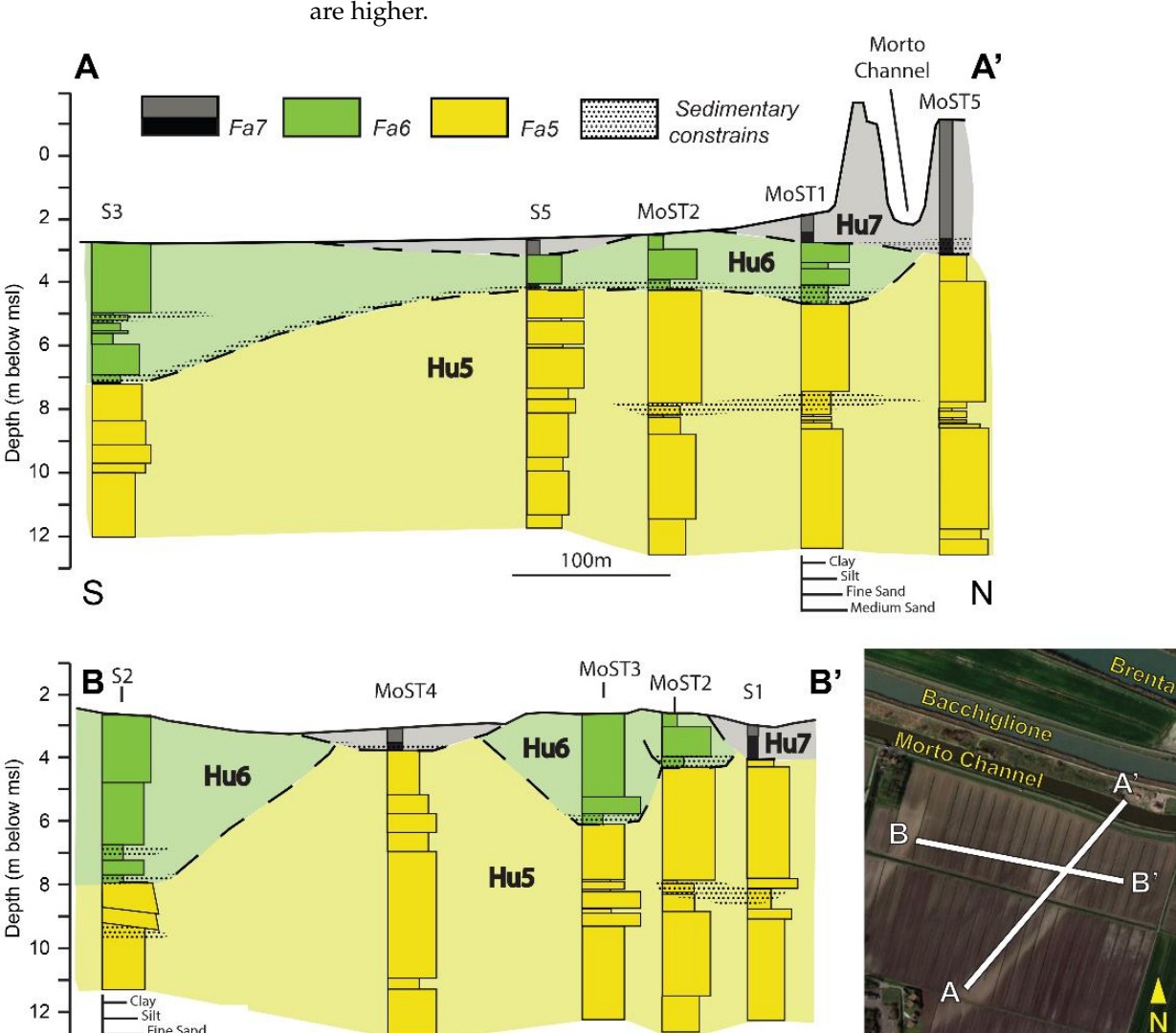

**Figure 14.** Stratigraphic sections of the upper portion of the subsoil, corresponding to the phreatic aquifer. The two sections cross the study area in roughly N-S and W-E directions (A-A′ and B-B′ in the inset map). In the sections, the stratigraphic constraints are indicated by dotted polygons.

5.3.2. W-E Section (Figure 15b)

- Best mitigation conditions: the eastern part of the area shows the highest EC levels (35–40 mS/cm, from bottom to the top in S1 well) and presents a fresher water lens in the MoST2 well. Moving to the west the surficial lens is present also in MoST3 well with EC values down to 15 ms/cm, where the deepest part reaches EC values of about 25–30 mS/cm. In the western part, the upper aquifer presents a freshwater lens extending for a width of about 10 m and a thickness of 7–8 m.
- Worst mitigation conditions: the surficial freshwater lenses disappear except for the western one that only decreases its overall extension.

The analyses of the interpolations allow to recognize different aquifer behaviors: in the western and southernmost portions of the phreatic aquifer (S2 and S3), a freshwater lens, presenting EC below 6 mS/cm, is always present, with variable thickness (between 5 and 8 m) depending on the availability of freshwater. In the central part of the area (MoST1, MoST2 and MoST3), a water lens with EC around 16 mS/cm develops in the upper aquifer

in periods with best mitigation conditions. In the other areas (MoST4, S1, S5), the aquifer is always salty, whereas it is always fresh in MoST5.

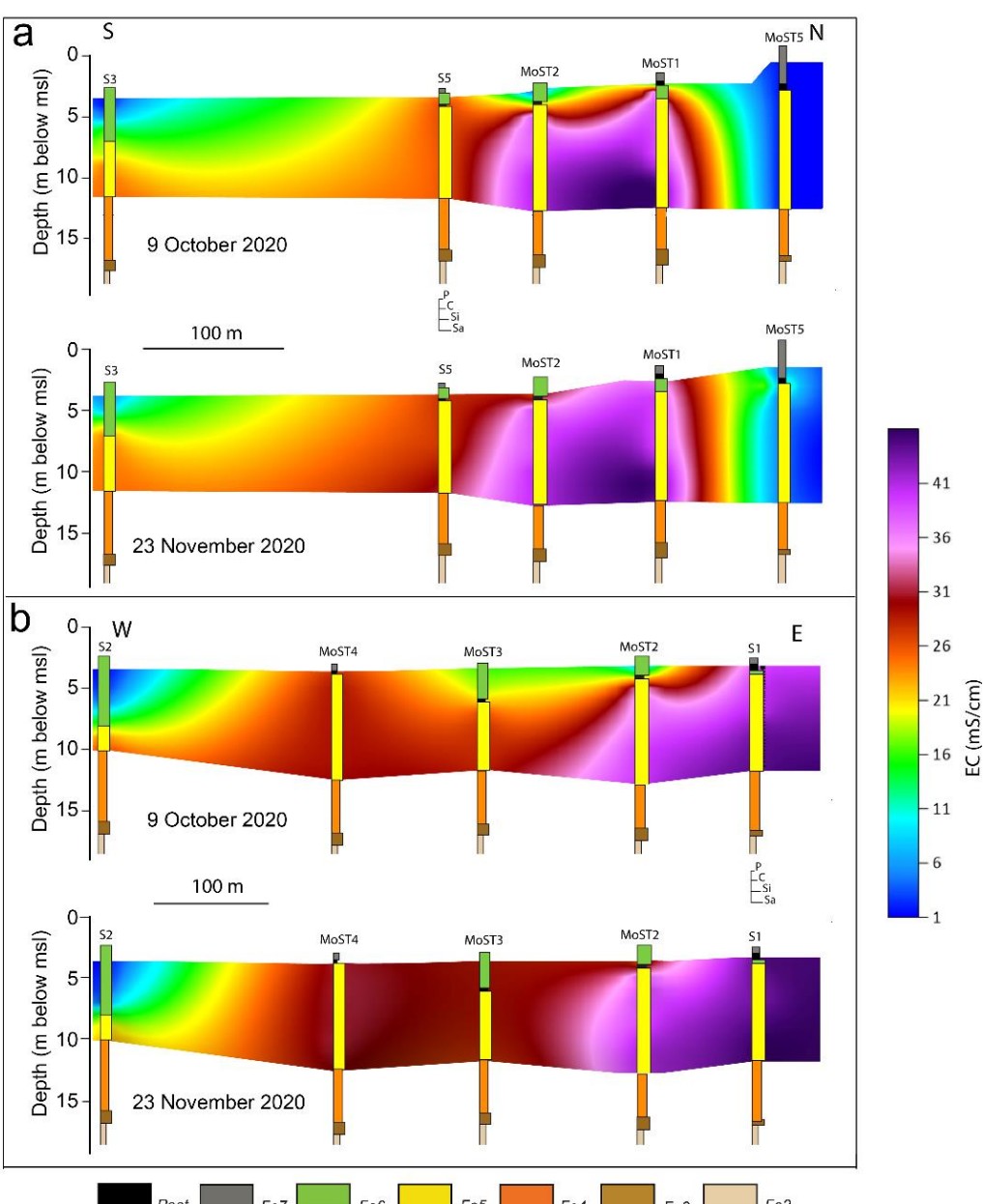

**Figure 15.** Interpolation of EC vertical profiles recorded on 9 October 2020 and 23 November 2020: (**a**) in N-S direction, (**b**) in W-E direction. The location of the stratigraphy of the studied cores and wells is reported in Figures 3 and 14.

The comparison between sections in Figure 16 and the stratigraphic columns in Figure 15 outlines the following. (i) the presence of Hu7 at the top of the subsoil prevents the formation of a freshwater lenses (MoST4, S1, S5). The clays and peat layers composing this unit prevent the freshwater infiltration and the subsequent mitigation of the aquifer salinity. The only exception is recorded in MoST1 site, where a fresher water lens develops in wet conditions. A possible explanation is the proximity of the freshwater inputs from the Morto Channel and the Brenta–Bacchiglione fluvial system: when the amount of precipitation is high the seepage/dispersion from the riverbeds increases, mitigating the nearby portion of the aquifer. (ii) the presence of Hu6 (S2, MoST3, MoST2, S3, MoST1) allows the formation of fresher water lenses in the upper part of the aquifer. (iii)

MoST5, being always fresh due to the proximity of the Morto Channel, could not outline stratigraphic influence on the groundwater dynamic. (iv) the fresher water lenses always rest on a sedimentologic constraint.

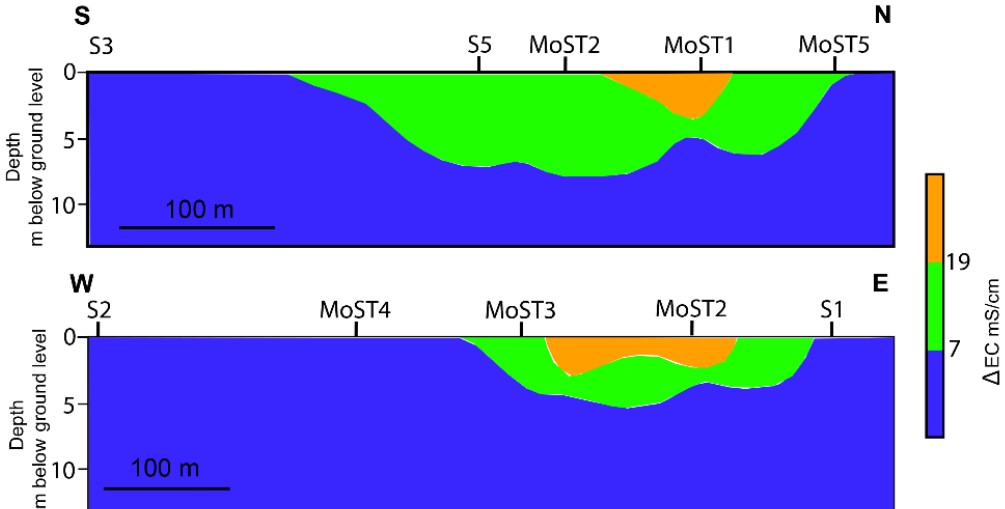

**Figure 16.** Interpolation of the EC difference between best and worst mitigation conditions in the N-S and W-E section shown in Figure 15. The location of the monitoring wells is reported.

Finally, differences of the EC values between the best and worst mitigation conditions in NS and WE sections (Figure 16) show that the major variability is located in the north-central part of the aquifer in correspondence to MoST1, MoST2 and MoST3 wells. In the southernmost and westernmost sectors, the variability is low because the freshwater lens is always present, due to the presence of a well-developed channelized aquifer (Hu6), reaching 4–5 m below the ground level. In the northernmost sector, the aquifer is always fresh because of the seepage inputs from Morto Channel, while in the easternmost it is always salty. In conclusion, the area of the aquifer in which the availability of freshwater influences more the salinity, is the north-central part of the aquifer, where paleochannels up to 3 m correspond to the Hu6 unit.

## 6. Conclusions

In this study, hydrogeological and sedimentological analyses were used to investigate the influence of stratigraphic discontinuities in shaping different behaviors of the salinity stratification into the unconfined aquifer at the southern margin of the Venice lagoon (Italy).

The main outcomes can be summarized as follows:

- We demonstrated that the groundwater dynamics is influenced by sedimentological constraints.
- The constraints correspond to: peat layers at the base of Hu7, fine-grained layers inside Hu6 or at its base, and the transition between finer and coarser sand in Hu3.
- In the phreatic aquifer we recognized three possible groundwater dynamics conditions: (i) in the westernmost and southernmost portions a thick freshwater lens is always present. The lenses are confined at the top of the aquifer by stratigraphic constraints located at different depths, depending on the amount of precipitation. This condition occurs where the Hu6 is thicker and Hu7 is absent; (ii) in the central-northern part of the area, a fresher water lens could develop in the most surficial part when the availability of freshwater allows the formation of the lens (MoST1, MoST2, MoST3). The lenses are confined at the base by the stratigraphic constraint at the base of Hu6. This condition occurs where the Hu6 is present and Hu7 is absent (except for MoST1 that possibly receives freshwater inputs from the Morto Channel); (iii) in the other areas, where Hu7 is present and Hu6 is absent (S1, S5, MoST4, MoST5), freshwater

lenses never develop, and the aquifer is always salty (except for the MoST5 site that receives freshwater from the Morto Channel and the aquifer is always fresh). In these areas, the stratigraphic constraints are barely recognizable and separate waters with similar EC.

- The portion of the aquifer most influenced by freshwater availability is the north-central part, where a paleochannel system (Hu6) is present, but with minor thickness than at the westernmost and southernmost part of the area. This area, being the most sensitive to changes in freshwater inputs, should be considered in the framework of eventual mitigation strategies.
- We demonstrated the importance of a detailed reconstruction of the subsoil architecture in order to understand the stratigraphic influence on the fresh–saltwater dynamics into the aquifers. Thus, a detailed sedimentological analysis is critical to optimize water management in highly salinized aquifer systems.
- A detailed analysis of the geological—geomorphological variations resulting from the Holocene evolution of the coastal zones, along with hydrogeological analyses, are basic/fundamental tools for the exploration of freshwater lenses in coastal areas.

This study represents a considerable advancement in the understanding of fresh–saltwater dynamics into the aquifers of the lowlands at the southern margin of the Venetian lagoon. Yet, a number of uncertainties still need to be solved by further investigations, such as those related to the stratigraphic porosity [51] and permeability, pressure and period of the tide, especially to address optimal strategies for mitigation and water-management practices.

**Author Contributions:** Conceptualization, C.C., A.B. and L.T.; methodology, C.C., A.B. and L.T.; software, C.C., A.B. and L.T.; validation, M.C., C.D.L., S.D., C.T. and L.Z.; formal analysis, C.C., A.B. and L.T.; investigation, C.C., S.D. and L.T.; data curation, C.C., A.B. and L.T.; writing—original draft preparation, C.C.; writing—review and editing, A.B., M.C., C.D.L., S.D., C.T., L.T. and L.Z.; supervision, L.T.; project administration, L.T.; funding acquisition, S.D. and L.T. All authors have read and agreed to the published version of the manuscript.

**Funding:** This research was funded by the contribution from the EU co-financing and the Interreg Italy–Croatia CBC Programme 2014–2020 (Priority Axes: Safety and Resilience) through the European Regional Development Fund as a part of the project "Monitoring seawater intrusion in coastal aquifers and Testing pilot projects for its mitigation (MoST)" (AID: 10047743). This work was also supported by the Scientific Cooperation Program between the National Research Council of Italy and the Chinese Academy of Sciences (Bilateral Agreement CNR-CAS), 2020–2022 Joint Research Project "Coastal system changes over the Anthropocene: Natural Vs Induced drivers in China and Italy".

**Institutional Review Board Statement:** Not applicable.

**Informed Consent Statement:** Not applicable.

**Data Availability Statement:** Datasets are available upon request.

**Conflicts of Interest:** The authors declare no conflict of interest.

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
