# Peer review of "Morpho-Sedimentary Constraints in the Groundwater Dynamics of Low-Lying Coastal Area: The Southern Margin of the Venice Lagoon, Italy"

_water, doi:10.3390/w14172717_

Round 1

Reviewer 1 Report

The paper should be accepted.

More suggestions can be found in the attachment. 

Author Response

We would like to thank the reviewers for their comments and suggestions. We think that their help has been extremely important to improve our work. We hope to have addressed all the comments and we ask revisors, if eventually some comments were not clear for us, to explain them again to us in their next revision

Reviewer 1

The text is written like a report, not like a research paper. I recommend a native speaker proof-reading

we hope to have improved the style of the paper, thanks to your comments. If it still written like a report please let us now in your future revision and we will try to improve it more

Line 18: please avoid inaccurate terms as often

done

General comment: concentrate more on results and their interpretation in the abstract

We described with more details our results in the abstract, adding the following: “Furthermore, our analyses reveal that the differences in stratigraphic architecture of the upper 10 meters of the subsoil result in different behavior of the phreatic aquifer in respect to the saltwater intrusion. In particular our results outline three possible configuration of the fresh-saltwater dynamics into the aquifer: i) where the subsurface is characterized by the presence of a thick, up to 5 meters, paleo-channel, it always contains a fresh-water lens in the most surficial part of the phreatic aquifer; ii) where the subsurface is composed by fine sediments of marsh and lagoon paleo-environment, the phreatic aquifer tends to be salt-contaminated from the top to the bottom; iii) where the subsurface contains thin, up to 2-3 meters, paleo-channel deposits, the fresh-saltwater dynamics of the most surficial part of the phreatic aquifer varies more during the year, as a result of seasonal variation in precipitations.”

Line 31: remove Venice.

Done

Line 34: do you mean decreased/declined

We meant “increased”. We corrected it

Line 35: long drought instead of longe-drought

Done

Line 46: what kind of the intrusion?

Saltwater intrusion. Done

Lines 52-53: which detailed analyses was adopted?

We added: using facies analyses method

Line 55: “of floodplain, alluvial fan and delta “instead of “as floodplains and alluvial fans and

Delta”

Done

Line 56: “several studies” or “some studies”?

We correct it in “some studies”

Line 60: please remove comma

done

Lines 60-64: why do we need this information?

Lines 62- (through remote sensing, field surveys, Vertical Electrical

Soundings, sediment tests from boreholes and EC measurements) too detailed.

Done

Lines 65-68: single-sentence paragraphs should be avoided. Please re-write this sentence.

We have merged the paragraph to the previous one

Line 78: “intend” or “Understand”

The correct word is “to mean”. We corrected it

Line 81-103: avoid brackets.

done

Lines 83-84: avoid brackets.

done

Lines 82-86: single-sentence paragraphs should be avoided. Please re-write this sentence.

We have merged the paragraph to the previous one

Lines 87-93: it is methodology.

We deleted this part

Line 103-117: in this area instead of in the area

done

Line 105: what does it intrude irregularly?

The saline plume. We specified it.

Line 106: top of?

The top of the saline plume. We think that, having repeated it in the previous sentence, it should be clear now.

Lines 97-: specify Figure 1: a, b, c or d.

done

Line 130: This period = late Pleistocene and Holocene.

done

Line 137-187: whose base dates back to 8-10 Kyr BP on what basis is it dated?

It is based on Tosi et al. 2007 (geological map of the area). We put the reference in the right place in the text

Line 144: you use Fig. and Figure, please unify it in the whole

text.

done

Lines 151-153: I do not understand this sentence.

There are 3 sedimentary bodies composing the aquifer of the area. The aquifer a is composed by Pleistocene sandy deposits and it represents a confined aquifer lying below the Caranto paleosoil. Aquifers b and c represent together the phreatic aquifer. Aquifer b is composed by sandy deposits extending all along the area. Aquifers c are isolated and smaller aquifers confined in paleo-channels.

Aquifer instead of aquifer (with capital letter)

Done

Modified instead of mod.

Done

Line 160: why do not ‘deltaic and alluvial deposits’,’lagoon deposits’, ‘lower-shoreface deposits’, ‘caranto’ discharge as aquifers?

Because these deposits are composed by fine sediments with low permeability, representing aquiclude and aquitarde. It is better explained in the following paragraph (facies associations and hydrostratigraphic units)

Line 175 (257): why some cores are MoST and some S? Because of locality? It should be

explained.

This is because the cores have been done in the frame of two different projects. We think that explaining it here it sounds too “report style” and we preferred to not put this detail in the paper

Line 176: do not list examples, list them all

done

Line 178: see Error! Reference source not found. what does it mean?

It was an error of the program. We solved it. Now it is “see Fig.2”

Line 179: you mention Figure 3, but you didn’t mentioned Figure 2 until now

Now Fig. 2 is mentions before fig. 3

Line 180: “different” please explain what kinf of HU

We do not think, as for the FA, that this is the correct chapter to listed the kind of HU. It is all explain in the chapter “facies associations and hydrostratigraphic units”

Line 183: Error! Reference source not found. the same as in line 178.

Solved it

Line 185: Figure 3 contains 3 parts: S-N logs, W-E logs and the map. Use the map as a

separate Figure (Figure 2).

We are sorry… we do not understand this suggestion. We think that the figure is clear and more complete with the map inside it

Lines 175-187: please avoid single-sentence paragraphs. I recommend to link them in the one

paragraph.

done

Line 191: from which June to which July?

June 2020 to june 2021. It is written in the text

Line 194: (m on the asl)???

Meters on the average sea level. But we substituted it with msl to be more homogeneous with the other part of the text

Line 198: asl is it above sea level? Clarify it.

It is average sea level. Now we put msl

Lines 197-200: please link this paragraph with the previous one.

done

Line 202: (lithology, granulometry, sorting) ref. Pszonka et al., 2021.

done

Line: 211 (344): numbers from 0 to 10 in numerals, numbers above in words therefore 7 instead

done

Line 212: why do you start with the Fa5 to 7? What about earlier ones Fa-1 to 4?

It is not a list, it is just a sentence to make readers noticing that the phreatic aquifer, that is the focus of the work, correspond to the upper fa (fa5 to fa 7) We decided to remove it. It is clear in the following list of FA

Lines 213-215: please remove this sentence. It is the article, no a report.

done

Line 216: Facies association 1 do not exceed 3 m in thickness. It is only in this case. The same

as next Fas 2-7.

We are sorry we do not understand this comment… please explain better what you would like to change or clarify. Facies association 1 is the lowest one. For that reason we do not sea the base of this FA and we can only say its minimum thickness.

Line 220: information or data?

Data. Done

Line 233: confining at the top Hu1 it is obvious because you wrote earlier that Fa2

overlies Fa1.

When we talk in term of Hu we are talking from an hydrological perspective, for that reason we think that is important to outline that the Caranto paleosol represent the confinement of the aquifer (Hu1)

Line 236: structureless, and poorly sorted. Remember this is not a report,

you write the scientific paper.

We regularly find terms like“structurless and poorly sorted” in sedimentological descriptions on scientific papers. We do not think that it sounds like a report

Line 257: please name the aquifer of facies associations 1-4.

Done

Line 259: please do not write every time FaX is found above FaXX. It is obvious. Write it once at the very beginning and do not repeat the obvious information.

When it was possible we deleted the repetitive information

Line 265: is interpreted by whom? References.

Tosi et al., 2007. We put the reference

Lines 277-281: is it your interpretation or someone earlier wrote it? Please provide references.

Fabbri et al., 2013. We added the reference

Line 295: please indicate cores 1-5 and 7 in the measured section as it is done for FA 5 and 6.

We are sorry we do not understand this comment. All the FA are indicated in the measured sections

You use Fa in the text and FA in the figure. Please unify it

Done

Lines 302-311: one-sentence paragraphs. I recommend to avoid them

Unified it with the next ones

Lines 307- Please avoid sentences like: “A summary of… is reported”

We changed this part, putting the reference to the table in brackets (See Table 1 for the correlation between FA, HU and seismic and sequence stratigraphic units defined in previous studies)

Lines 308-309: Error! Reference source not found..

We corrected the problem

Line 320: “in this section” avoid such phrases, it for reports.

Done

Groundwater dynamics this part I s out of my competences, however I have a few questions and suggestions: Please clarify which core belongs to W-E sections, which to N-S sections.

We think this information is already clear in the text: 4.3.1 describes the W-E section and 4.3.2 the N-S. The cores described in each section belong to the section specified in the subtitle

Why you describe S2, then MoST4, MoST3, etc. Why this order?

In 4.3.1 the order is from West to East. In 4.3.2 the order is from North to south. It becomes clear looking to the figure 3 and 15

Please avoid “are described in the follows”

done

Please avoid “only”, “quite”, “indeed”

done

Show from the bottom to the top

We are sorry we do not understand this comment

You still use the word “show”. Sometimes replace them with indicate, display or present

done

Lines 514-516: it is a speculation.

We deleted it

Line 538: do you mean “the aquifer allows to…”?

Yes, we corrected it

Line 547: avoid to add some information in brackets, in the whole text.

done

Lines 554-557: it is a methodology.

Line 558: report style.

It is true. We deleted it and rewrited. See the next lines

Lines 560 -571: I recommend rewrite these sentences

We re-wrote the text trying to be more clear: “Five setting where stratigraphic constraints occur have been recognized in two orthogonal sections crossing the study area (Figure 15),:

At the base of Hu6, where clay and silty layers are present. The presence of the channelized aquifers trigger the formation and the maintenance through time of fresher water lenses in the phreatic aquifer. This occurs homogeneously along the S-N section, corresponding to a N-S directed paleochannel, and in correspondance of MoST2, Most3 and S2 points, in the W-E section.

At the top of clay and silty layers inside Hu6. This occurs in the western channelized surficial sandbody (observed in S2 point) and in the southern part of the eastern one (observed in S3 point), where the channelized aquifers (Hu6) show their major thickness.

Inside the littoral aquifer Hu5, in the upper part, where coarser sand passes downward to finer one (as in S2 point).

Around the middle portion of Hu5, where silty-clay layers appear (in S1, MoST1 and MoST2 points)

At the base of Hu7, in correspondence to peaty clay layers that could preserve fresher water at the very top of the phreatic aquifer (as it was observed in MoST5, and MoST4 points).

Line 572: why is Hu6 the most important?

We decided not to consider one constrain more important than the others. It is too subjective. We deleted this part

Lines 582-586: it is a methodology.

It is true. We moved it in the right section

Lines 589, 599, 604, 611: why are some words bold?

We put them in normal characters

Lines 592- “The top of the aquifer is characterized by the presence of a fresher water lens up to 1-2 m” instead of “The top of the aquifer in this area is characterized by the presence of a fresher water lens up to 1-2 m”

We cannot change it, because not all the aquifer is characterized by the fresh water lens, but only in that particular area

Avoi to start a sentence with “moving to”

done

Line 611: the main difference between what?

Between worst and best mitigation conditions. We change the sentence to: Worst mitigation conditions: the surficial freshwater lenses disappear except for the western one, only decreasing its overall extension.

Lines 631-632: Error! Reference source not found.

Corrected it with the reference to the fig. 15

Line 63 avoid: we can state the following

We revised it as follows:: “The comparison between these sections and the stratigraphic sections in Figure 15 outlines that”

Lines 636-643: it is really hard to read.

It is important to read it looking at fig. 15 and fig. 16. We think that looking at the figures (as indicated in the text) it is important to understand this part… then the text should be result easy to read

647: please remove “indeed”

done

Line 667: what the constraints? Could be or are?

“are”, we corrected it

Line 688: how do you demonstrate?

We demonstrate it in fig. 17, showing the variation in EC between best and worst mitigation conditions (i.e. availability of surficial freshwaters). The area that varies more in EC is the north-central one, where the channelized aquifer is present but it is thinner. The westernmost and southernmost portion of the study site, where the channelized aquifer is thicker, always present a freshwater lens. While the other areas are always salty, meaning that even in humid conditions the aquifer remain salty from the top to the bottom. For that reasons we can say that the northerncentral part of the area is the most sensitive to freshwater supply from precipitation, and eventually, better responds to mitigation measures as for example sub-irrigation systems.

Reviewer 2 Report

Please find attached my comments.

Yours sincerely

Author Response

We would like to thank the reviewers for their comments and suggestions. We think that their help has been extremely important to improve our work. We hope to have addressed all the comments and we ask revisors, if eventually some comments were not clear for us, to explain them again to us in their next revision

Reviewer 2

Physical parameters related to formation and silt should be clarified, such as porosity and permeability coefficient, etc. It is hoped that the author can supplement some data in this respect.

Thank you for this suggestion. Unfortunately, we do not have data of porosity and permeability coefficient. The cores have been only analysed in terms of facies analyses and we do not have them anymore. But we are planning future studies considering these parameters.

In the last paragraph, Line 697, the relevant reference needs to be added: Although this study represents a step forward in understanding the fresh-saltwater dynamics into the aquifers, we are aware that a number of uncertainties still need to be resolved by further investigations, such as related to stratigraphic structure, stratigraphic porosity [1], the pressure and period of the ocean waves, and so on. References: [1] Sun Jichao, Huang Yuefei. Modeling the Simultaneous Effects of Particle Size and Porosity in Simulating Geo-Materials. Materials, 2022, 15(4): 1576.

done

“EC” first appears in Line63, But what is the EC? And Line 190 introduces Electroconductivity (EC) Should explain EC in Line 63.

Thank you. We deleted the part in which appeared for the first time EC. Now the first appearance is in line 190.

3. The author can put the monitoring data of 9 points (only part of the data) on a map for a spatial description, which may or may not be better.

We are sorry we do not understand this comment. Please explain it to us in your future revision

                                                                               4. Whether wave pressure, groundwater pressure are monitored synchronously? Did the study test particle gradation in different strata? Authors can add such data. If you have difficulty, you can add a statement like this sentence in introduction: The change of salinity concentration is directly related to the stratigraphic structure, stratigraphic porosity and particle gradation[1]. This paper only tests the electrical conductivity.

Unfortunately, we do not possess these data. We added the final statement as you suggested.

Round 2

Reviewer 2 Report

Authors have taken into account reviewers' suggestions, significantly enriched the references, and rewritten the text from the report form to a scientific paper, which was my main comment. Therefore, I believe that the article can be published in the present form.

As a non-native English speaker, I do not feel qualified to judge about the English language and style.

Author Response

Thank you.